# Tight nanoscale clustering of Fcγ receptors using DNA origami promotes phagocytosis

**Nadja Kern[1,2], Rui Dong[1,2], Shawn M Douglas[1], Ronald D Vale[1,2,3]\*, Meghan A Morrissey[1,4]\***

[1]Department of Cellular and Molecular Pharmacology, University of California San Francisco, San Francisco, United States; [2]Howard Hughes Medical Institute, University of California San Francisco, San Francisco, United States; [3]Howard Hughes Medical Institute Janelia Research Campus, Ashburn, United States; [4]Department of Molecular, Cellular and Developmental Biology, University of California Santa Barbara, Santa Barbara, United States

**Abstract** Macrophages destroy pathogens and diseased cells through Fcγ receptor (FcγR)-driven phagocytosis of antibody-opsonized targets. Phagocytosis requires activation of multiple FcγRs, but the mechanism controlling the threshold for response is unclear. We developed a DNA origami-based engulfment system that allows precise nanoscale control of the number and spacing of ligands. When the number of ligands remains constant, reducing ligand spacing from 17.5 nm to 7 nm potently enhances engulfment, primarily by increasing efficiency of the engulfment-initiation process. Tighter ligand clustering increases receptor phosphorylation, as well as proximal downstream signals. Increasing the number of signaling domains recruited to a single ligand-receptor complex was not sufficient to recapitulate this effect, indicating that clustering of multiple receptors is required. Our results suggest that macrophages use information about local ligand densities to make critical engulfment decisions, which has implications for the mechanism of antibody-mediated phagocytosis and the design of immunotherapies.

**\*For correspondence:**
Ron.Vale@ucsf.edu (RDV);
morrissey@ucsb.edu (MAM)

**Competing interests:** The authors declare that no competing interests exist.

## Introduction

Immune cells eliminate pathogens and diseased cells while limiting damage to healthy cells. Macrophages, professional phagocytes and key effectors of the innate immune system, play an important role in this process by engulfing opsonized targets bearing 'eat me' signals. One of the most common 'eat me' signals is the immunoglobulin G (IgG) antibody, which can bind foreign proteins on infected cells or pathogens. IgG is recognized by Fcγ receptors (FcγR) in macrophages that drive antibody-dependent cellular phagocytosis (ADCP) (*DiLillo et al., 2014*; *Erwig and Gow, 2016*; *Nimmerjahn and Ravetch, 2008*). ADCP is a key mechanism of action for several cancer immunotherapies including rituximab, trastuzumab, and cetuximab (*Chao et al., 2010*; *Uchida et al., 2004*; *Watanabe et al., 1999*; *Weiskopf et al., 2013*; *Weiskopf and Weissman, 2015*). Exploring the design parameters of effective antibodies could provide valuable insight into the molecular mechanisms driving ADCP.

Activation of multiple FcγRs is required for a macrophage to engulf a three-dimensional target. FcγR-IgG must be present across the entire target to drive progressive closure of the phagocytic cup that surrounds the target (*Griffin et al., 1975*). In addition, a critical antibody threshold across an entire target dictates an all-or-none engulfment response by the macrophage (*Zhang et al., 2010*). Although the mechanism of this thresholded response remains unclear, receptor clustering plays a role in regulating digital responses in other immune cells (*Berger et al., 2020*; *Davis and*

**eLife digest** The word 'phagocytosis' means cellular eating. It is the process by which cells extend their membranes around foreign particles and engulf them. Macrophages, a type of immune cell found in every tissue of the body, perform phagocytosis to eat pathogens and diseased cells. To avoid eating healthy cells, macrophages focus on targets marked by proteins called antibodies. They look for cells coated with high levels of a type of antibody called immunoglobulin G, or IgG for short, but only eat cells coated with enough IgG, raising the question, can macrophages count?

Macrophages recognize IgG antibodies using cell surface receptors called Fc-gamma Receptors. When these receptors bind to IgG, they cluster together. Researchers do not yet know how the number of IgG antibodies per cluster, or the spacing between them, affects phagocytosis. To find this out, researchers need to be able to manipulate the clustering experimentally. One way to do this is using a technique called DNA origami. This technique creates nanoscale patterns of DNA strands on a target surface. If the part of a receptor that interacts with its target is then replaced with a complementary DNA strand to the strands on the target surface, the receptor will bind the surface following the nanoscale pattern. This allows researchers to generate synthetic targets with specific patterns of receptor-target interaction.

Kern et al. replaced the part of the macrophage Fc-gamma Receptor that interacts with IgG with a strand of DNA. They then used DNA origami to arrange complementary DNA strands on pegboards and attached these pegboards to silica beads. The different arrangements of DNA on these pegboards mimicked the types of antibody clusters macrophages might encounter on the surfaces of the cells and particles they have to engulf in the body. Kern et al. found that tight clusters of the DNA targets on the pegboards made the macrophages most likely to begin phagocytosis, particularly clusters of eight or more DNA strands spaced less than seven nanometers apart. Macrophages encountering these tight clusters showed an increase in Fc-gamma receptor activation, which is crucial for macrophage attack.

Whether or not macrophages can count, they can at least sense the level of clustering of IgG antibodies to determine if a target should be engulfed. Doctors use antibody therapies that rely on Fc-gamma receptor engagement to treat cancer, autoimmune and neurodegenerative diseases. Understanding how clustering affects phagocytosis could aid in the design of new antibody treatments. It could also help improve the design of synthetic receptors to create designer immune cells that can attack specific targets. The next step will be to recreate the results from the synthetic system used by Kern et al. with natural receptors and antibodies.

van der Merwe, 2006; Holowka and Baird, 1996; Kato et al., 2020; Ma et al., 2020; Veneziano et al., 2020). FcγR clustering may also regulate phagocytosis (Goodridge et al., 2012). High-resolution imaging of macrophages has demonstrated that IgG-bound FcγRs form clusters (resolution of >100 nm) within the plasma membrane (Lin et al., 2016; Lopes et al., 2017; Sobota et al., 2005). These small clusters, which recruit downstream effector proteins such as Syk-kinase and phosphoinositide 3-kinase, eventually coalesce into larger micron-scale patches as they migrate towards the center of the cell-target synapse (Jaumouillé et al., 2014; Lin et al., 2016; Lopes et al., 2017; Sobota et al., 2005).

Prior observational studies could not decouple ligand clustering from other parameters, such as ligand number or receptor mobility. As a result, we do not have a clear picture of how ligand number or molecular spacing regulates signal activation. To directly assess such questions, we have developed a reconstituted system that utilizes DNA origami to manipulate ligand patterns on a single-molecule level with nanometer resolution. We found that tightly spaced ligands strongly enhanced phagocytosis compared to the same number of more dispersed ligands. Through manipulating the number and spacing of ligands on individual origami pegboards, we found that eight or more ligands per cluster maximized FcγR-driven engulfment, and that macrophages preferentially engulfed targets that had receptor-ligand clusters spaced ≤7 nm apart. We demonstrated that tight ligand clustering enhanced receptor phosphorylation, and the generation of PIP$_3$ and actin filaments – critical downstream signaling molecules – at the phagocytic synapse. Together, our results suggest that the nanoscale clustering of receptors may allow macrophages to discriminate between

lower density background stimuli and the higher density of ligands on opsonized targets. These results have implications for the design of immunotherapies that involve manipulating FcγR-driven engulfment.

## Results

### Developing a DNA-based chimeric antigen receptor to study phagocytosis

To study how isolated biochemical and biophysical ligand parameters affect engulfment, we sought to develop a well-defined and tunable engulfment system. Our lab previously developed a synthetic T cell signaling system, in which we replaced the receptor-ligand interaction (TCR-pMHC) with complimentary DNA oligos (*Taylor et al., 2017*). We applied a similar DNA-based synthetic chimeric antigen receptor to study engulfment signaling in macrophages. In our DNA-CARγ receptor, we replaced the native extracellular ligand-binding domain of the FcγR with an extracellular SNAP-tag that covalently binds a benzyl-guanine-labeled single-stranded DNA (ssDNA) (receptor DNA; *Figure 1a*; *Morrissey et al., 2018*). The SNAP-tag was then joined to the CD86 transmembrane domain followed by the intracellular signaling domain of the FcRγ chain (*Nimmerjahn and Ravetch, 2008*). We expressed the DNA-CARγ in the macrophage-like cell line RAW264.7 and the monocyte-like cell line THP-1.

As an engulfment target, we used silica beads coated with a supported lipid bilayer to mimic the surface of a target cell. The beads were functionalized with biotinylated ssDNA (ligand DNA) containing a sequence complementary to the receptor DNA via biotin-neutravidin interactions (*Figure 1A*). We used a ligand DNA strand that has 13 complementary base pairs to the receptor DNA, which we chose because the receptor-ligand dwell time (~24 s; *Taylor et al., 2017*) was comparable to the dwell time of IgG-FcγR interactions (~30–150 s; *Li et al., 2007*).

To test whether this synthetic system can drive specific engulfment of ligand-functionalized silica beads, we used confocal microscopy to measure the number of beads that were engulfed by each cell (*Figure 1B, C*). The DNA-CARγ drove specific engulfment of DNA-bound beads in both RAW264.7 and THP-1 cells (*Figure 1C*, *Figure 1—figure supplement 1*). The extent of engulfment was similar to IgG-coated beads, and the ligand density required for robust phagocytosis was also comparable to IgG [*Figure 1D*, *Figure 1—figure supplement 1*; *Bakalar et al., 2018*; *Morrissey et al., 2020*]. As a control, we tested a variant of the DNA-CAR that lacked the intracellular domain of the FcRγ chain (DNA-CAR_adhesion). Cells expressing the DNA-CAR_adhesion failed to induce engulfment of DNA-functionalized beads (*Figure 1C*), demonstrating that this process depends upon the signaling domain of the FcγR. Together, these data show that the DNA-CARγ can drive engulfment of targets in a ligand- and FcγR-specific manner.

### DNA origami pegboards activate DNA-CARγ macrophages

DNA origami technology provides the ability to easily build three-dimensional objects that present ssDNA oligonucleotides with defined nanometer-level spatial organization (*Hong et al., 2017*; *Rothemund, 2006*; *Seeman, 2010*; *Shaw et al., 2019*; *Veneziano et al., 2020*). We used DNA origami to manipulate the spatial distribution of DNA-CARγ ligands in order to determine how nanoscale ligand spacing affects engulfment. We used a recently developed two-tiered DNA origami pegboard that encompasses a total of 72 ssDNA positions spaced 7 nm and 3.5 nm apart in the x and y dimensions, respectively (*Dong et al., 2021*; *Figure 2A*, *Figure 2—figure supplement 1*). Each of the 72 ligand positions can be manipulated independently, allowing for full control over the ligand at each position (*Figure 2—figure supplement 1*). The DNA origami pegboard also contains fluorophores at each of its four corners to allow for visualization, and 12 biotin-modified oligos on the bottom half of the pegboard to attach it to a neutravidin-containing supported lipid bilayer or glass coverslip (*Figure 2A, B*, *Figure 2—figure supplement 1*).

To determine if the DNA origami pegboards could successfully activate signaling, we first tested whether receptors were recruited to the origami pegboard in a ligand-dependent manner. Using TIRF microscopy, we quantified the fluorescence intensity of the recruited GFP-tagged DNA-CARγ receptor to origami pegboards presenting 0, 2, 4, 16, 36, or 72 ligands (*Figure 2B–E*). Using signal from the 72 ligand (72L) origami pegboard as an internal intensity standard of brightness, and thus

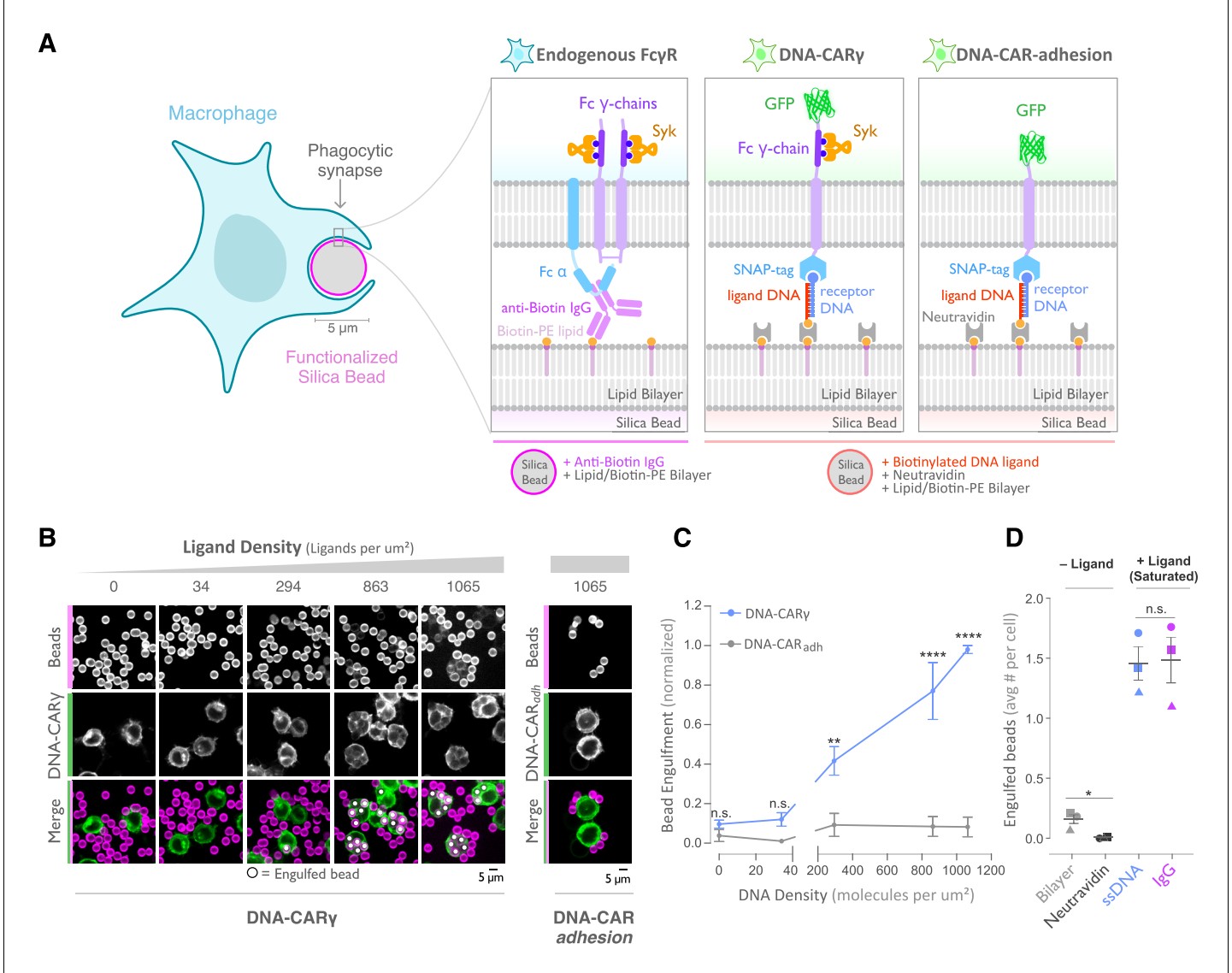

**Figure 1.** A DNA-based system for controlling engulfment. (**A**) Schematic shows the endogenous (left box) and DNA-based (middle and right boxes) engulfment systems. Engulfment via endogenous FcγRs (left box) is induced through anti-biotin IgG bound to 1-oleoyl-2-(12-biotinyl (aminododecanoyl))-sn-glycero-3-phosphoethanolamine (biotin-PE) lipids incorporated into the bilayer surrounding the silica bead targets. Engulfment induced via the DNA-based system uses chimeric antigen receptors (CAR) expressed in the macrophage and biotinylated ligand DNA that is bound to the lipid bilayer surrounding the silica bead. The DNA-CARγ (middle box) consists of a single-stranded DNA (ssDNA) (receptor DNA) covalently attached to an extracellular SNAP-tag fused to a CD86 transmembrane domain, the intracellular domain of the FcRγ chain, and a fluorescent tag. The DNA-CAR$_{adhesion}$ (right box) is identical but lacks the signaling FcRγ chain. (**B**) Example images depicting the engulfment assay. Silica beads were coated with a supported lipid bilayer (magenta) and functionalized with neutravidin and the indicated density of ligand DNA (*Figure 1—figure supplement 1A*). The functionalized beads were added to RAW264.7 macrophages expressing either the DNA-CARγ or the DNA-CAR$_{adhesion}$ (green) and fixed after 45 min. The average number of beads engulfed per macrophage was assessed by confocal microscopy. Scale bar denotes 5 μm here and in all subsequent figures. Internalized beads are denoted with a white sphere in the merged images. (**C**) The number of beads engulfed per cell for DNA-CARγ (blue) or DNA-CAR$_{adhesion}$ (gray) macrophages was normalized to the maximum bead eating observed in each replicate. Dots and error bars denote the mean ± SEM of three independent replicates (n ≥ 100 cells analyzed per experiment). (**D**) DNA-CARγ-expressing macrophages were incubated with bilayer-coated beads (gray) functionalized with anti-biotin IgG (magenta), neutravidin (black), or neutravidin and saturating amounts of ssDNA (blue). The average number of beads engulfed per cell was assessed. Full data representing the fraction of macrophages engulfing specific numbers of IgG or ssDNA beads is shown in *Figure 1—figure supplement 1*. Each data point represents the mean of an independent experiment, denoted by symbol shape, and bars denote the mean ± SEM. n.s. denotes p>0.05, * indicates p<0.05, ** indicates p<0.005, and **** indicates p<0.0001 by a multiple t-test comparison corrected for multiple comparisons using the Holm–Sidak's method (**C**) or Student's t-test (**D**).
The online version of this article includes the following figure supplement(s) for figure 1:

*Figure 1 continued on next page*

*Figure 1 continued*

**Figure supplement 1.** DNA-based engulfment system reflects endogenous engulfment.

correcting for differences in illumination between wells, we found that the average fluorescence intensity correlated with the number of ligands presented by individual origami pegboards (*Figure 2D, E*). In addition, we measured Syk recruitment to individual DNA origami pegboards and found that Syk intensity also increased as a function of the number of ligands present on each origami pegboard (*Figure 2C*, *Figure 2—figure supplement 2*). These results confirmed that our DNA origami system provides a platform that allows quantitative receptor recruitment and the analysis of downstream signaling pathways.

## Nanoscale clustering of ligand enhances phagocytosis

FcγR cluster upon ligand binding, but the functional importance of such clustering for phagocytosis has not been directly addressed, and whether a critical density of receptor-ligand pairs is necessary to initiate FcγR signaling is unclear (*Duchemin et al., 1994*; *Jaumouillé et al., 2014*; *Lin et al., 2016*; *Lopes et al., 2017*; *Sobota et al., 2005*). To address these questions, we varied the size of ligand clusters by designing DNA origami pegboards presenting 2–36 ligands. To ensure a constant total number of ligands and origami pegboards on each bead, we mixed the signaling origami pegboards with 0-ligand 'blank' origami pegboards in appropriate ratios (*Figure 3A*). We confirmed that the surface concentration of origami pegboards on the beads was comparable using fluorescence microscopy (*Figure 3—figure supplement 1*). We found that increasing the number of ligands per cluster increased engulfment, but that engulfment plateaued at a cluster size of eight ligands (*Figure 3b*). We confirmed that the observed engulfment phenotype was both ligand, receptor, and FcγR signaling dependent (*Figure 3C, D*). Together, these data reveal that FcγR clustering strongly enhances engulfment up to a cluster size of eight ligands.

## Spatial organization of ligands in nanoclusters regulates engulfment

Next, we examined whether distance between individual receptor-ligand molecules within a signaling cluster impacts engulfment. For this experiment, we varied the spacing of four ligands on the origami pegboard. The 4-ligand tight origami (4T) contains four ligands clustered at the center of the pegboard (7 nm by 3.5 nm square), the medium origami (4M) has ligands spaced 21 nm by 17.5 nm apart, and the spread origami (4S) has four ligands positioned at the four corners of the pegboard (35 nm by 38.5 nm square) (*Figure 4A*). We found that the efficiency of macrophage engulfment was approximately twofold higher for the 4T-functionalized beads when compared to the 4M or 4S beads (*Figure 4A*). We confirmed via fluorescence microscopy that the concentration of origami pegboards on the surface was similar, and therefore ligand numbers on the beads were similar (*Figure 4—figure supplement 1*). Human THP-1 cells expressing the DNA-CARγ showed the same ligand spacing dependence (*Figure 4—figure supplement 1*). In addition, we generated DNA-CAR constructs containing the FcRγ and α chain transmembrane domains that would be present in the endogenous receptor complex (*Figure 4—figure supplement 1*). To minimize dimerization between FcRγ transmembrane domains, we either made a C25Aγ chain mutation, as this cysteine forms a disulfide bridge between γ chains, or truncated the transmembrane domain before this residue. We found that the efficiency of macrophage engulfment was dependent on ligand spacing for all constructs tested (*Figure 4—figure supplement 1*). Expression of the various DNA-CARs at the cell cortex was comparable, and engulfment of beads functionalized with both the 4T and the 4S origami platforms was dependent on the FcγR signaling domain (*Figure 4—figure supplement 1*). Together, these results demonstrate that macrophages preferentially engulf targets with tighter ligand clusters.

Tightly spaced ligands could potentially increase phagocytosis by enhancing the avidity of receptor-ligand interactions within each cluster. Such a hypothesis would predict that tightly spaced ligands increase DNA-CARγ-BFP occupancy at the phagocytic cup. However, when we measured the total fluorescence intensity of receptors at the phagocytic cup, we did not detect a difference in DNA-CARγ-BFP recruitment to 4T and 4S beads (Figure 6A, B). However, to eliminate any potential contribution of avidity, we created 4T and 4S origami pegboards with very high-affinity 16mer DNA

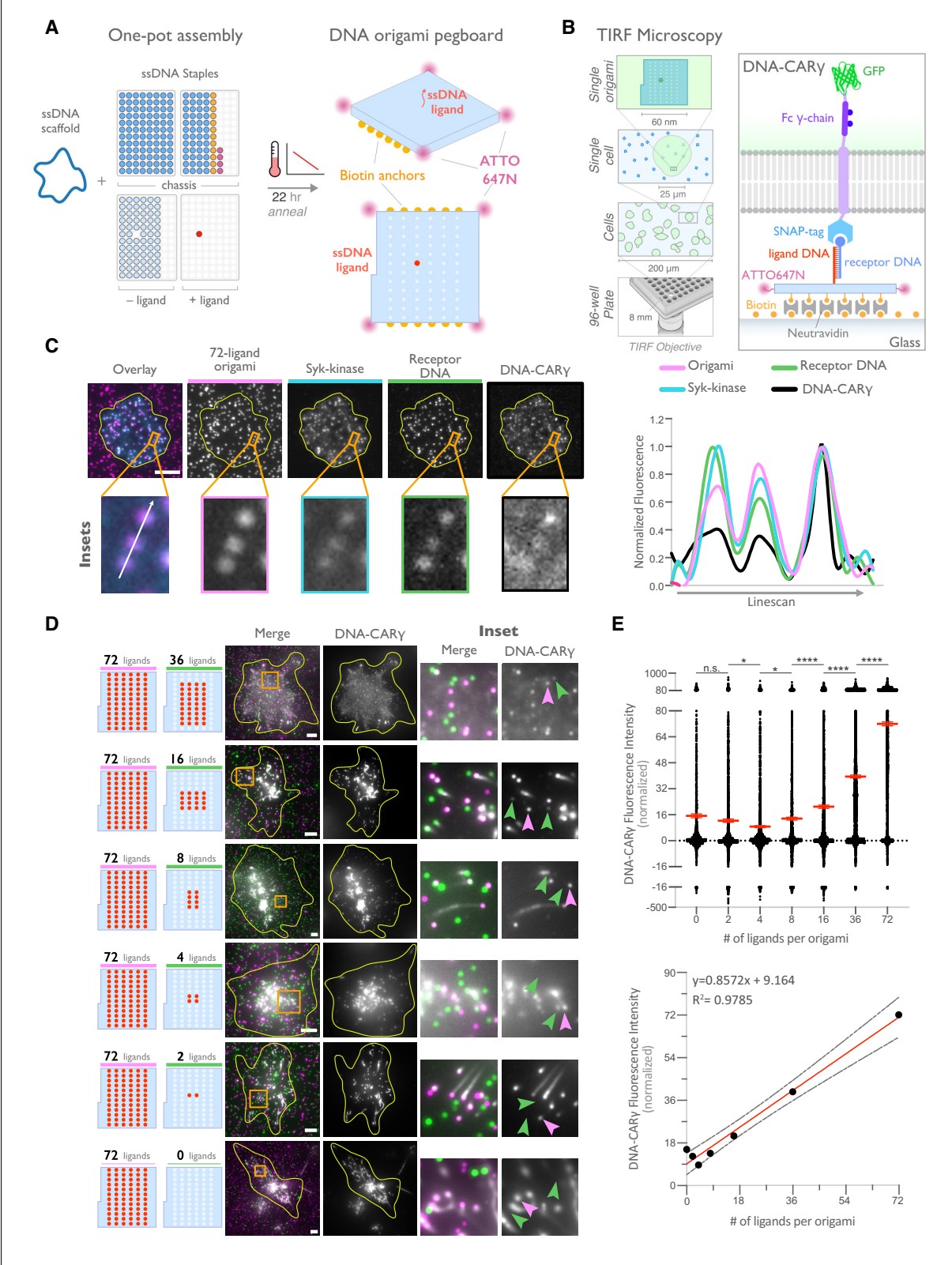

**Figure 2.** DNA origami pegboard induces ligand-dependent signaling. (**A**) Schematic shows the DNA origami pegboard used in this study (right) and the components used to create it using a one-pot assembly method (left, *Figure 2—figure supplement 1*). The top of the two-tiered DNA origami pegboard has 72 positions spaced 7 nm and 3.5 nm apart in the x and y dimensions, which can be modified to expose a single-stranded ligand DNA (red) or no ligand (light blue). A fluorophore is attached at each corner of the pegboard for visualization (pink). The bottom tier of the pegboard

*Figure 2 continued on next page*

*Figure 2 continued*

displays 12 biotin molecules (yellow) used to attach the origami to neutravidin-coated surfaces. Full representation of the DNA origami pegboard assembly is shown in *Figure 2—figure supplement 1*. (B) Schematic portraying the Total Internal Reflection Fluorescence (TIRF) microscopy setup used to image THP-1 cells interacting with origami pegboards functionalized to glass coverslips in (C) and (D) (left). On the right is a zoomed-in side view of an origami pegboard functionalized to a biotin (yellow) and neutravidin (gray) functionalized glass coverslip and interacting with a single DNA-CARγ receptor. (C) TIRF microscopy images of THP-1 cells show that the DNA-CARγ (BFP; fifth panel; black in linescan), the receptor DNA bound to the DNA-CARγ (Cy5; fourth panel; green in linescan), and Syk (mNeonGreen; third panel; cyan in merge and linescan) are recruited to individual 72-ligand (72L) origami pegboards (Atto-647; second panel; magenta in merge and linescan). Each diffraction-limited magenta spot represents an origami pegboard. The top panels show a single cell (outlined in yellow), and the bottom insets (orange box in top image) show three origami pegboards at higher magnification. The linescan (right, area denoted with a white arrow in merged inset) shows the fluorescence intensity of each of these channels. Intensity was normalized so that 1 is the highest observed intensity and 0 is background for each channel. (D) TIRF microscopy images show DNA-CARγ-expressing THP1s interacting with 72L origami pegboards (pink) and origami pegboards presenting the indicated number of ligands (pegboards labeled in green). Left schematics represent origami pegboard setups for each row of images where red dots denote the presence of a ligand DNA. Middle images depict a single macrophage (outlined in yellow), and right images show the area indicated with an orange box on the left. Examples of DNA-CARγ-mNeonGreen (gray) recruitment to individual origami pegboards is marked by pink (72L origami pegboard) and green (origami pegboard with the indicated ligand number) arrowheads (right). (E) Quantification of experiment shown in (D). Top graph shows the DNA-CARγ intensity at the indicated origami pegboard type normalized to the average DNA-CARγ intensity at 72L origami pegboards in the same well. Each dot represents one origami pegboard, and red lines denote the mean ± SEM of pooled data from three separate replicates. n.s. denotes $p > 0.05$, * indicates $p < 0.05$, and **** indicates $p < 0.0001$ by an ordinary one-way ANOVA with Holm–Sidak's multiple comparison test. A linear regression fit (bottom) of the average fluorescence intensities of each of the origami pegboards suggests that the mean DNA-CARγ fluorescent intensities are linearly proportional to the number of ligands per DNA origami pegboard. The black dots represent the mean normalized DNA-CARγ intensity, the red line denotes the linear regression fit, and the gray lines show the 95% confidence intervals.

The online version of this article includes the following source data and figure supplement(s) for figure 2:

**Source data 1.** Receptor raw intensities.
**Figure supplement 1.** Design and assembly of nanoscale ligand-patterning pegboard built from DNA origami.
**Figure supplement 2.** Syk intensity increases with ligand number in origami cluster.
**Figure supplement 2—source data 1.** Syk raw intensities.

ligands that are predicted to dissociate on a time scale of >7 hr (*Taylor et al., 2017*; *Figure 4B*). Using these 16mer high-affinity ligands, we found that 4T origami beads were still preferentially engulfed over 4M or 4S origami beads (*Figure 4B*, *Figure 4—figure supplement 1*). These results suggest that an avidity effect is not the cause of the preferential engulfment of targets having tightly spaced ligands.

## Tight ligand spacing enhances engulfment initiation and downstream signaling

We next determined how ligand spacing affects the kinetics of engulfment. Using data from live-cell imaging, we subdivided the engulfment process into three steps: bead binding, engulfment initiation, and engulfment completion (*Figure 5A*, *Video 1*). To compare engulfment dynamics mediated by 4T and 4S origami pegboards in the same experiment, we labeled each pegboard type with a different colored fluorophore, functionalized a set of beads with each type of pegboard, and added both bead types to macrophages at the same time (*Figure 5B*, *Video 2*). Macrophages interacted with beads functionalized with the 4T and 4S pegboards with comparable frequency (46 ± 7% total bead-cell contacts vs. 54 ± 7% total bead-cell contacts, respectively). However, the probability of engulfment initiation was significantly higher for the 4T (95 ± 5% of bead contacts) versus 4S (61 ± 9% of bead contacts) beads, and the probability that initiation events resulted in successful completion of engulfment was higher for 4T (69 ± 9% of initiation events) versus 4S (39 ± 11% of initiation events) beads (*Figure 5A*). Initiation events that failed to induce successful engulfment either stalled after progressing partially over the bead or retracted the extended membrane back to the base of the bead. In addition, for beads that were engulfed, the time from contact to engulfment initiation was ~300 s longer for beads functionalized with 4S origami pegboards than beads containing 4T origami pegboards (*Figure 5C*). However, once initiated, the time from initiation to completion of engulfment did not differ significantly for beads coated with 4T or 4S origami (*Figure 5D*). Overall, 66 ± 8% of 4T bead contacts resulted in successful engulfment compared to 24 ± 8% for 4S beads (*Figure 5E*). The DNA-CAR~adhesion~ macrophages rarely met the initiation criteria, suggesting that active signaling from the FcγR is required (*Figure 5—figure supplement 1*). Together, these data

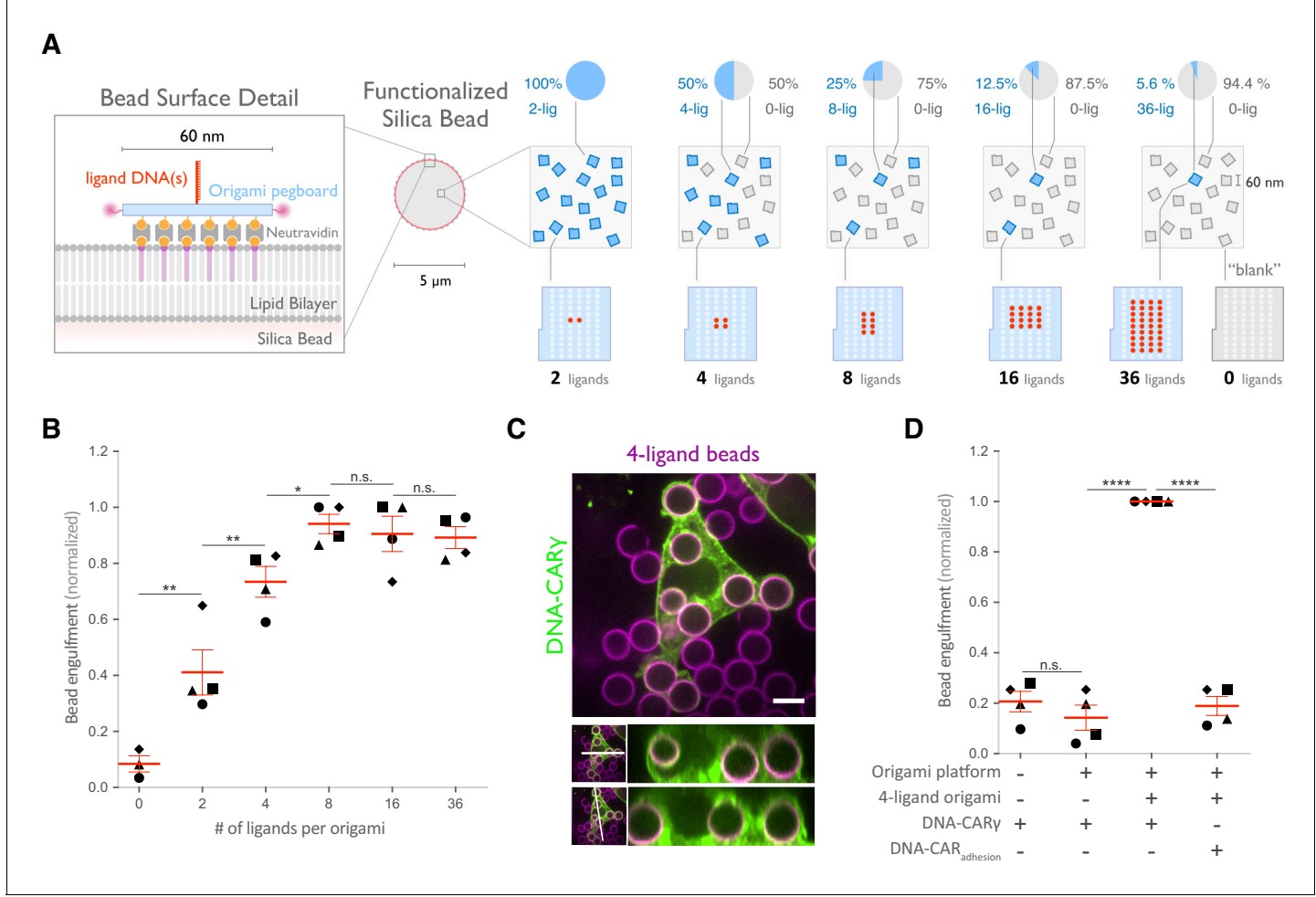

**Figure 3.** Nanoscale clustering of ligand enhances phagocytosis. (A) Schematic showing an origami pegboard functionalized to a lipid bilayer surrounding a silica bead (left) and the origami pegboard mixtures used to functionalize the bilayer-coated silica beads for experiment quantified in (B) (right). Blue squares represent origami pegboards with the indicated number of ligands (schematics below, red dot denotes ligand DNA and light blue dot denotes no ligand), and gray squares represent 0-ligand 'blank' origami pegboards. Pie charts above describe the ratios of ligand origami presenting pegboards to 'blank' pegboards. (B) Beads were functionalized with mixtures of origami pegboards containing the indicated ligand-presenting origami pegboard and the 0-ligand 'blank' origami pegboards in amounts designated in (A). The graph depicts the number of beads internalized per DNA-CARγ-expressing macrophage normalized to the maximum bead eating in that replicate. Each dot represents an independent replicate (n ≥ 100 cells analyzed per experiment), denoted by symbol shape, with red lines denoting mean ± SEM. Data is normalized to the maximum bead eating in each replicate. (C) Example image showing the DNA-CARγ (green) drives engulfment of beads (bilayer labeled in magenta) functionalized with 4-ligand DNA origami pegboards. A cross section of the z plane indicated in the inset panel (white line, bottom) shows that beads are fully internalized. (D) Bilayer-coated silica beads were functionalized with neutravidin, neutravidin and DNA origami pegboards presenting 0 DNA ligands, or neutravidin and 4-ligand DNA origami pegboards. The graph depicts normalized bead eating per cell of the indicated bead type for cells expressing the DNA-CARγ or the DNA-CAR$_{adhesion}$. Each dot represents an independent replicate, denoted by symbol shape (n ≥ 100 cells analyzed per experiment), with red lines denoting mean ± SEM. The data are normalized to the maximum bead eating in each replicate. * denotes p<0.05, ** denotes p<0.005, **** denotes p<0.0001, and n.s. denotes p>0.05 in (B) and (D) as determined by an ordinary one-way ANOVA with Holm–Sidak's multiple comparison test.

The online version of this article includes the following figure supplement(s) for figure 3:

**Figure supplement 1.** Origami intensity on beads is comparable across conditions.

reveal that tighter spacing between ligands within a cluster enhances the probability and kinetics of initiating engulfment, as well as the overall success frequency of completing engulfment, but does not affect the rate of phagosome closure once initiated.

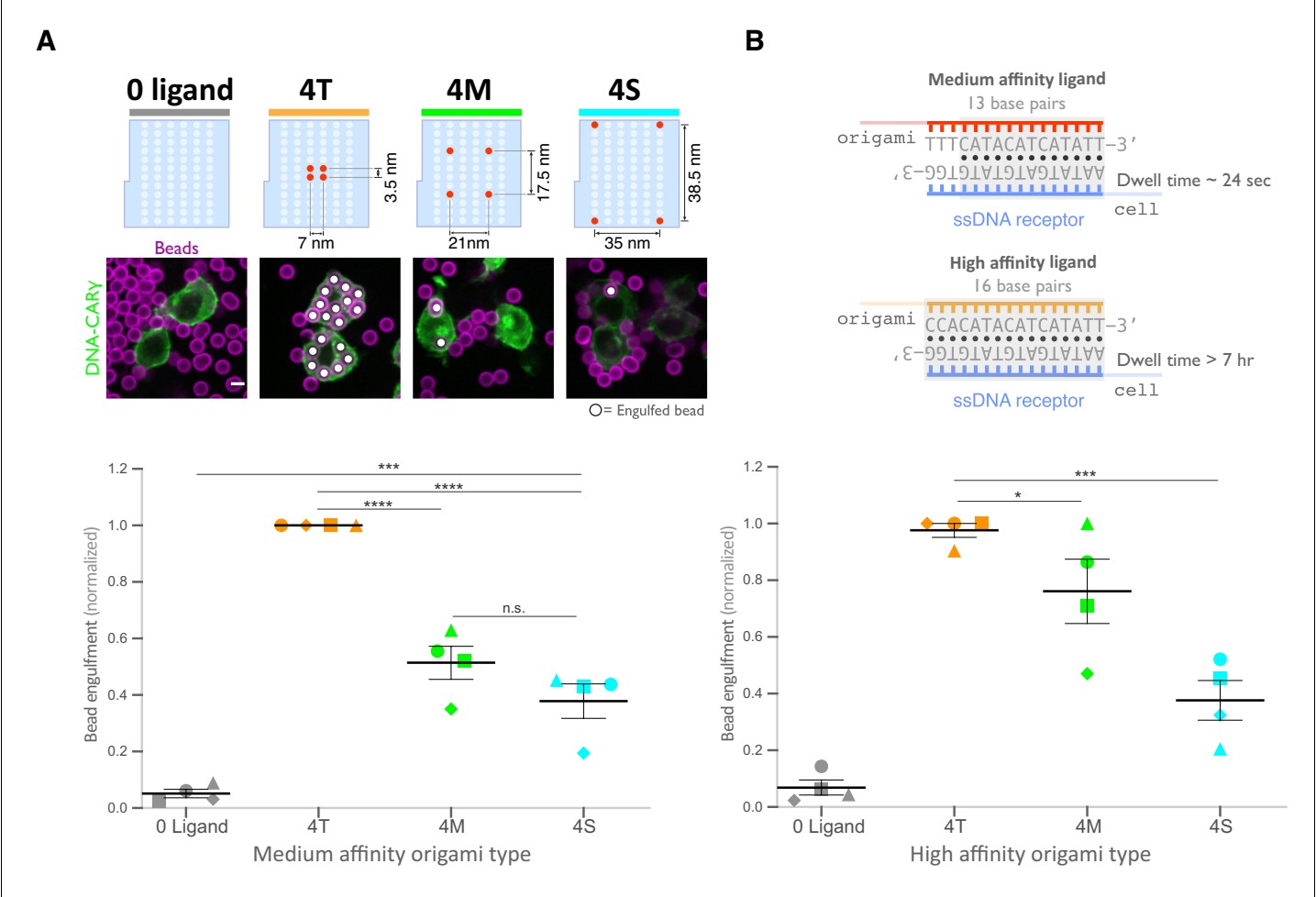

**Figure 4.** Spatial arrangement of ligands within nanoclusters regulates engulfment. (**A**) Schematics (top) depict 4-ligand origami pegboards presenting ligands at the positions indicated in red. Beads were functionalized with 0-ligand 'blank' (gray) origami pegboards, 4T (orange) origami pegboards, 4M (green) origami pegboards, or 4S (cyan) origami pegboards at equal amounts and fed to DNA-CARγ-expressing macrophages. Representative confocal images (middle) depict bead (bilayer in magenta) engulfment by macrophages (green). Internalized beads are denoted with a white sphere. Quantification of the engulfment assay is shown in the graph below depicting the number of beads engulfed per macrophage normalized to the maximum observed eating in that replicate. (**B**) Schematics of the receptor DNA (blue) paired with the medium-affinity 13 base pair DNA-ligand (red) used in all previous experiments including (**A**) and the high-affinity 16 base pair ligand-DNA (yellow) used for experiment shown in the graph below. Beads were functionalized with 0-ligand 'blank' (gray), high-affinity 4T (orange), high-affinity 4M (green), or high-affinity 4S (cyan) origami pegboards and fed to DNA-CARγ-expressing macrophages. Graph shows the number of beads engulfed per macrophage normalized to the maximum observed eating in that replicate. Each data point represents the mean of an independent experiment, shapes denote data from the same replicate, and bars show the mean ± SEM (**A, B**). * denotes p<0.05, *** denotes p<0.0005, **** denotes p<0.0001, and n.s. denotes p>0.05 as determined by an ordinary one-way ANOVA with Holm–Sidak's multiple comparison test (**A, B**).

The online version of this article includes the following figure supplement(s) for figure 4:

**Figure supplement 1.** Ligand clustering enhances engulfment in RAW macrophages expressing DNA-CARs with endogenous FcγR transmembrane domains and in THP1s.

## Tightly spaced ligands enhance receptor phosphorylation

We next determined how the 4T or 4S origami pegboards affect signaling downstream of FcγR binding by measuring fold enrichment at the phagocytic cup compared to the rest of the cortex of (1) a marker for receptor phosphorylation (the tandem SH2 domains of Syk; *Bakalar et al., 2018*; *Morrissey et al., 2018*), (2) $PIP_3$ (via recruitment of the $PIP_3$ binding protein Akt-PH-GFP), and (3) filamentous actin (measured by rhodamine-phalloidin binding, *Figure 6A, B*). We found that 4T phagocytic cups recruited more tSH2-Syk than the 4S beads, indicating an increase in receptor phosphorylation by nanoclustered ligands. Generation of $PIP_3$ and actin filaments at the phagocytic cup

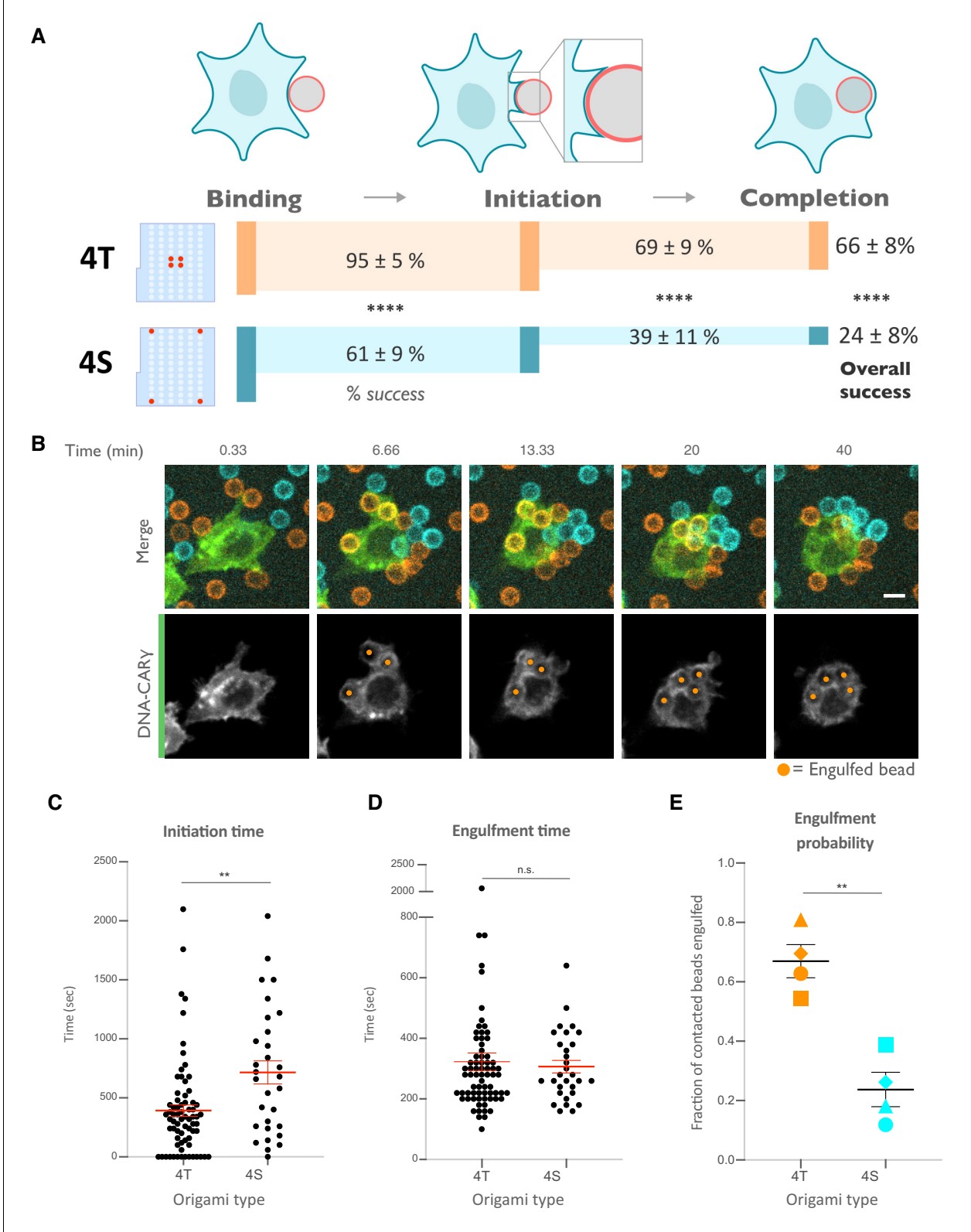

**Figure 5.** Nanoscale ligand clustering controls engulfment initiation. (**A**) Schematic portraying origami pegboards used to analyze the steps in the engulfment process quantified in (**C–E**). Bead binding is defined as the first frame the macrophage contacts a bead; initiation is the first frame in which the macrophage membrane has begun to extend around the bead, and completion is defined as full internalization. The macrophage membrane was visualized using the DNA-CARγ, which was present throughout the cell cortex. The % of beads that progress to the next stage of engulfment (%

*Figure 5 continued on next page*

*Figure 5 continued*

success) is indicated for 4T (orange, origami labeled with Atto550N) and 4S (cyan, origami labeled with Atto647N) beads. **** denotes p<0.0001 as determined by Fisher's exact test. (**B**) Still images from a confocal microscopy timelapse showing the macrophage (green) interacting with both the 4T origami pegboard-functionalized beads (orange) and the 4S origami pegboard-functionalized beads (cyan), but preferentially engulfing the 4T origami pegboard-functionalized beads. In the bottom panel (DNA-CARγ channel), engulfed beads have been indicated by a sphere colored to match its corresponding origami type. (**C**) Graph depicts quantification of the time from bead contact to engulfment initiation for all beads that were successfully engulfed. Each dot represents one bead with red lines denoting mean ± SEM. (**D**) Graph depicts the time from engulfment initiation to completion. Each dot represents one bead with red lines denoting mean ± SEM. (**E**) Graph shows the fraction of contacted 4T and 4S beads engulfed (orange and cyan, respectively) by the macrophages. Data represent quantification from four independent experiments, denoted by symbol shape, and bars denote the mean ± SEM. n.s. denotes p>0.05 and ** indicates p<0.005 by Student's t-test comparing the 4T- and 4S-functionalized beads (**C–E**).

The online version of this article includes the following figure supplement(s) for figure 5:

**Figure supplement 1.** DNA-CAR_adhesion fails to induce frequent engulfment initiation attempts.

also increased at 4T relative to 4S synapses (*Figure 6B*). This differential recruitment of downstream signaling molecules to 4T versus 4S origami beads was most apparent in early and mid-stage phagocytic cups; late-stage cups showed only a slightly significant difference in tSH2-Syk recruitment and no significant differences in generation of PIP$_3$ or actin filaments (*Figure 6—figure supplement 1*). Together, these data demonstrate that nanoscale ligand spacing affects early downstream signaling events involved in phagocytic cup formation.

We next sought to understand why distributing ligands into tight clusters enhanced receptor phosphorylation and engulfment. One possibility is that the clustering of four complete receptors is needed to drive segregation of the inhibitory phosphatase CD45 and allow sustained phosphorylation of the FcγR immune receptor tyrosine-based activation motif (ITAM) (*Bakalar et al., 2018*; *Freeman et al., 2016*; *Goodridge et al., 2012*; *Schmid et al., 2016*). Alternatively, the 4-ligand cluster may be needed to obtain a critical intracellular concentration of FcγR ITAM signaling domains. To test for the latter possibility, we designed a synthetic receptor (DNA-CAR-4xγ) that contains four repeats of the intracellular domain of the DNA-CARγ connected by a GGSG linker between each repeat (*Figure 6C*). We confirmed that this DNA-CAR-4xγ receptor was more potent in activating engulfment than an equivalent receptor (DNA-CAR-1xγ−3xΔITAM) in which the three C-terminal ITAM domains were mutated to phenylalanines (*Figure 6C, D*). Keeping the number of intracellular ITAMs constant, we compared the engulfment efficiency mediated by two different receptors: (1) the DNA-CAR-4xγ that interacted with beads functionalized with 1-ligand origami and (2) the DNA-CAR-1xγ−3xΔITAM that interacted with beads coated with equivalent amounts of 4T origami (*Figure 6C*). While the DNA-CAR-1xγ−3xΔITAM-expressing macrophages engulfed 4T origami beads, the DNA-CAR-4xγ macrophages failed to engulf the high-affinity 1-ligand origami beads (*Figure 6D*, *Figure 6—figure supplement 1*). To ensure that all four ITAM domains on the DNA-CAR-4xγ were signaling competent, we designed two additional DNA-CARs that placed the functional ITAM at the second and fourth position (*Figure 6—figure supplement 1*). These receptors were able to induce phagocytosis of 4T origami beads, indicating that the DNA-CAR-4xγ likely contains four functional ITAMs. Collectively, these results indicate that the tight clustering of multiple receptors is necessary for engulfment and increasing the number of intracellular signaling modules on a single receptor is not sufficient to surpass the threshold for activation of engulfment.

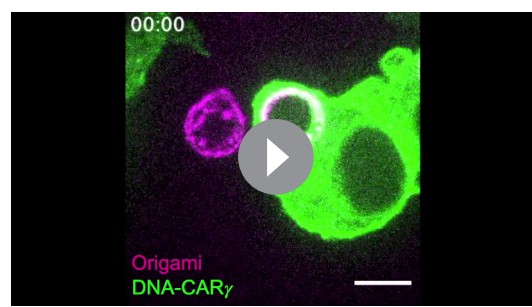

**Video 1.** The engulfment program broken into three steps: bead binding, engulfment initiation, and engulfment completion. A macrophage infected with the DNA-CARγ (green) engulfs a 5 µm silica bead coated in a supported lipid bilayer (magenta) and functionalized with 4T origami pegboards. The movie is a maximum intensity projection of z-planes and depicts the bead binding, initiation, and completion steps of the engulfment process. Time is indicated at the top left, and scale bar denotes 5 µm.

https://elifesciences.org/articles/68311#video1

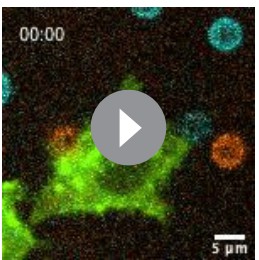

**Video 2.** DNA-CARγ macrophages preferentially engulf beads functionalized with tightly spaced ligands. A DNA-CARγ-expressing macrophage (green) interacts with 4T origami pegboard-functionalized beads (orange) and 4S origami pegboard-functionalized beads (cyan) that were added simultaneously and in equal amounts to the well of cells. The macrophage engulfs only 4T origami pegboard-functionalized beads. The movie is a maximum intensity projection of z-planes acquired every 20 s for 28 min. Time is indicated at the top left.

https://elifesciences.org/articles/68311#video2

## Discussion

Macrophages integrate information from many FcγR-antibody interactions to discriminate between highly opsonized targets and background signal from soluble antibody or sparsely opsonized targets. How the macrophage integrates signals from multiple FcγR binding events to make an all-or-none engulfment response is not clear. Here, we use DNA origami nanostructures to manipulate and assess how the nanoscale spatial organization of receptor-ligand interactions modulates FcγR signaling and the engulfment process. We found that tight ligand clustering increases the probability of initiating phagocytosis by enhancing FcγR phosphorylation.

Phagocytosis requires IgG across the entire target surface to initiate local receptor activation and to 'zipper' close the phagocytic cup (*Freeman et al., 2016*; *Griffin et al., 1975*). Consistent with this zipper model, incomplete opsonization of a target surface, or micron-scale spaces between IgG patches, decreases engulfment (*Freeman et al., 2016*; *Griffin et al., 1975*). Initially suggested as an alternative to the zipper model, the trigger model proposed that engulfment occurs once a threshold number of receptors interact with IgG (*Ben M'Barek et al., 2015*; *Griffin et al., 1975*; *Swanson and Baer, 1995*). While this model has largely fallen out of favor, more recent studies have found that a critical IgG threshold is needed to activate the final stages of phagocytosis (*Zhang et al., 2010*). Our data suggest that there may also be a nanoscale density-dependent trigger for receptor phosphorylation and downstream signaling. Taken together, these results suggest that both tight nanoscale IgG-FcγR clustering and a uniform distribution of IgG across the target are needed to direct signaling to 'zipper' close the phagocytic cup. Why might macrophages use this local density-dependent trigger to dictate engulfment responses? Macrophages constantly encounter background 'eat me' signals (*Gonzalez-Quintela et al., 2008*). This hyper-local density measurement may buffer macrophages against background stimuli and weakly opsonized targets that are unlikely to have adjacent bound antibodies, while still robustly detecting and efficiently engulfing highly opsonized targets.

Our findings are consistent with previous results demonstrating that FcγR crosslinking correlates with increased ITAM phosphorylation (*Huang et al., 1992*; *Kwiatkowska and Sobota, 2001*; *Lin et al., 2016*; *Sobota et al., 2005*). While our data pinpoints a role for ligand spacing in regulating receptor phosphorylation, it is possible that later steps in the phagocytic signaling pathway are also directly affected by ligand spacing. The mechanism by which dense-ligand clustering promotes receptor phosphorylation remains an open question, although our data rule out a couple of models. Specifically, we demonstrate that nanoscale ligand clustering does not noticeably affect the amount of ligand-bound receptor at the phagocytic cup, and that ligand spacing continues to affect engulfment when avidity effects are diminished through the use of high-affinity receptor-ligands. Collectively, these data reveal that changes in receptor binding or recruitment caused by increased avidity are unlikely to account for the increased potency of clustered ligands. Our data also exclude the possibility that receptor clustering simply increases the local intracellular concentration of FcγR signaling domains as arranging FcγR ITAMs in tandem did not have the same effect as clustering multiple receptor-ligand interactions. However, it remains possible that the geometry of the intracellular signaling domains could be important for activating or localizing downstream signaling, and that tandem ITAMs on the same polypeptide cannot produce the same engulfment signals as ITAMs on separate parallel polypeptides.

One possible model to explain the observed ligand-density dependence of signaling involves the ordering of lipids around the FcγR. Segregated liquid-ordered and liquid-disordered membrane

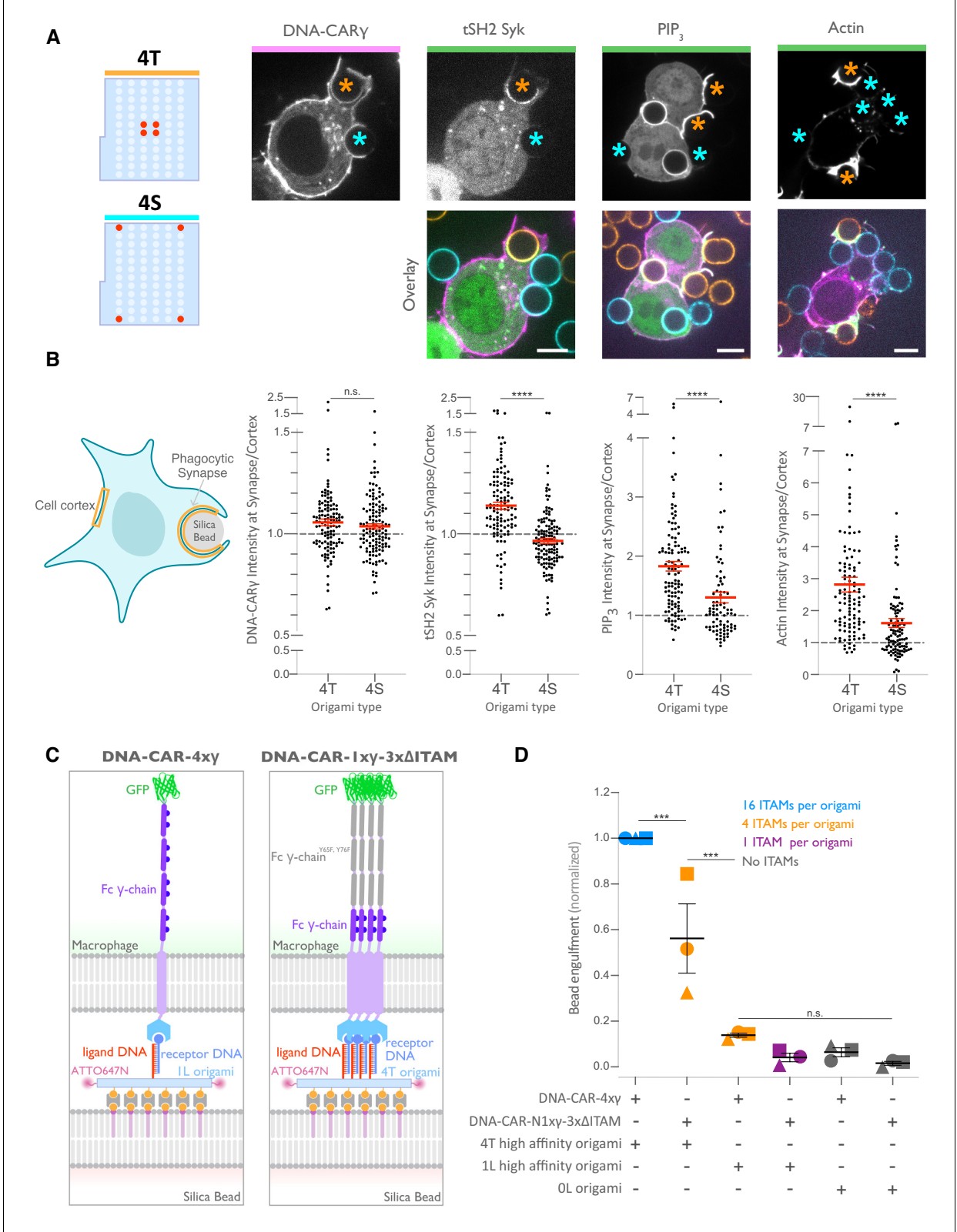

**Figure 6.** Nanoscale ligand spacing controls receptor activation. (**A**) Beads were functionalized with 4T (orange) or 4S (cyan) origami pegboards at equal amounts, added to macrophages expressing the DNA-CARγ (magenta) and the indicated signaling reporter protein (green; grayscale on top). Phagocytic synapses were imaged via confocal microscopy. Asterisks indicate whether a 4T (orange) or a 4S (cyan) bead is at the indicated phagocytic synapse in the upper panel. (**B**) Schematic (left) depicts the areas measured from images shown in (**A**) to quantify the fluorescence intensity
*Figure 6 continued on next page*

*Figure 6 continued*

(yellow outlines). Each phagocytic synapse measurement was normalized to the fluorescence intensity of the cell cortex at the same z-plane. Graphs (right) depict the ratio of fluorescence at 4T- or 4S-functionalized bead synapses to the cortex for the indicated reporter. Each dot represents one bead with red lines denoting mean ± SEM. (C) Schematic portraying the CAR constructs and origami used in the experiment quantified in (D). The DNA-CAR-4xγ construct (left) consists of four repeats of the intracellular domain of the DNA-CARγ connected by a GGSG linker. The DNA-CAR-1xγ−3xΔITAM (right) is identical to the DNA-CAR-4xγ except that the tyrosines composing the immune receptor tyrosine-based activation motif (ITAM) domains (purple circles) are mutated to phenylalanines in the three C-terminal repeats (gray). Cells expressing either of these constructs were fed beads functionalized with either high-affinity 1-ligand origami pegboards (left), high-affinity 4T origami pegboards (right), or 0-ligand 'blank' origami pegboards (not shown), and engulfment was assessed after 45 min. (D) Graph shows the number of beads engulfed per macrophage normalized to the maximum observed eating in that replicate. Each data point represents the mean from an independent experiment, denoted by symbol shape, and bars denote the mean ± SEM. Blue points represent a condition where 16 ITAMs are available per origami, orange points represent conditions where 4 ITAMs are available per origami, purple points represent a condition where 1 ITAM is available per origami, and gray points represent conditions where no ITAM is available. n.s. denotes $p > 0.05$, *** denotes $p < 0.0005$, and **** denotes $p < 0.00005$ as determined by the Student's t-test (B) or an ordinary one-way ANOVA with Holm–Sidak's multiple comparison test (D).

The online version of this article includes the following figure supplement(s) for figure 6:

**Figure supplement 1.** Differential recruitment of downstream signaling molecules is greater at early and mid-stage phagocytic cups.

domains around immune receptor clusters have been reported to promote receptor phosphorylation (*Bag et al., 2020*; *Dinic et al., 2015*; *Eggeling et al., 2009*; *Kabouridis, 2006*; *Simons and Ikonen, 1997*; *Sohn et al., 2006*; *Stone et al., 2017*). FcγR clusters are associated with liquid-ordered domains (*Beekman et al., 2008*; *Katsumata et al., 2001*; *Kwiatkowska and Sobota, 2001*). Liquid-ordered domains recruit Src family kinases, which phosphorylate FcγRs, while liquid-disordered domains are enriched in the transmembrane phosphatase CD45, which dephosphorylates FcγRs (*Bag et al., 2020*; *Sohn et al., 2006*; *Stone et al., 2017*). Thus, lipid ordering could provide a mechanism that leads to receptor activation if denser receptor-ligand clusters are more efficient in nucleating or associating with ordered lipid domains.

As an alternative model, a denser cluster of ligated receptors may enhance the steric exclusion of the bulky transmembrane proteins like the phosphatases CD45 and CD148 (*Bakalar et al., 2018*; *Goodridge et al., 2012*; *Zhu et al., 2008*). CD45 is heavily glycosylated, making the extracellular domain 25–40 nm tall (*Davis and van der Merwe, 2006*; *McCall et al., 1992*; *Woollett et al., 1985*). Because of its size, CD45 is excluded from close cell-cell contacts, such as those mediated by IgG-FcγR, which have a dimension of 11.5 nm (*Bakalar et al., 2018*; *Burroughs et al., 2011*; *Carbone et al., 2017*; *Chung et al., 2013*; *Lu et al., 2011*; *Schmid et al., 2016*). IgG bound to antigens ≤ 10.5 nm from the target surface induces CD45 exclusion and engulfment (estimated total intermembrane distance of ≤22 nm *Bakalar et al., 2018*). Our DNA origami structure is estimated to generate similar intermembrane spacing, consisting of hybridized receptor-ligand DNA (~9.4 nm), the origami pegboard (6 nm), and neutravidin (4 nm) (*Rosano et al., 1999*). A higher receptor-ligand density constrains membrane shape fluctuations (*Krobath et al., 2009*; *Krobath et al., 2011*; *Różycki et al., 2010*), and this constraint may increase CD45 exclusion (*Schmid et al., 2016*). Both the lipid ordering and the steric exclusion models predict at least a partial exclusion of the CD45 from the zone of the receptor cluster. However, the dimension of the tight cluster in particular is very small (7 by 3.5 nm) and measurement of protein concentration at this level is currently not easily achieved, even with super-resolution techniques. Overall, our results establish the molecular and spatial parameters necessary for FcγR activation and demonstrate that the spatial organization of IgG-FcγR interactions alone can affect engulfment decisions.

How does our synthetic DNA-CARγ receptor compare to endogenous FcγRs? Our DNA-CARs are single-chain receptors that recruit one intracellular signaling domain per ligand, similar to the single-chain human FcγRIIA receptor (*Nimmerjahn and Ravetch, 2006*). FcγRIIA is ubiquitously expressed on human myeloid cells, and high-affinity FcRIIA alleles correlate with an increase in effectiveness of the ADCP-inducing drug rituximab and lupus susceptibility (*Bruhns and Jönsson, 2015*; *Nimmerjahn and Ravetch, 2006*). The majority of FcγR family members, including all activating mouse FcγRs and the human FcγRI and FcγRIIIA, are multimeric complexes composed of a ligand-binding α chain and a dimerized signaling γ chain. This results in two signaling γ chains recruited to each IgG ligand. The different stoichiometry between ligand-binding and intracellular signaling domains may affect some parameters like optimal cluster size. A second difference between the

DNA-CARγ and the endogenous system is the presence of the CD86 transmembrane domain. We found that ligand spacing had a similar effect on phagocytosis when we replaced the CD86 transmembrane domain with the Fcα or Fcγ transmembrane domain. However, the Fcγ transmembrane domain construct triggered more bead internalization across all conditions. We hypothesize this could be because the transmembrane domain retains some ability to dimerize, recruiting more signaling domains to each ligand, or because it is better able to associate with lipid-ordered domains. Future studies that pattern either endogenous Fc receptor complex or IgG ligand could clarify these questions.

How does the spacing requirements for FcγR nanoclusters compare to other signaling systems? Engineered multivalent Fc oligomers revealed that IgE ligand geometry alters Fcε receptor signaling in mast cells (*Sil et al., 2007*). DNA origami nanoparticles and planar nanolithography arrays have previously examined optimal inter-ligand distance for the T cell receptor, B cell receptor, NK cell receptor CD16, death receptor Fas, and integrins (*Arnold et al., 2004*; *Berger et al., 2020*; *Cai et al., 2018*; *Deeg et al., 2013*; *Delcassian et al., 2013*; *Dong et al., 2021*; *Veneziano et al., 2020*). Some systems, like integrin-mediated cell adhesion, appear to have very discrete threshold requirements for ligand spacing while others, like T cell activation, appear to continuously improve with reduced intermolecular spacing (*Arnold et al., 2004*; *Cai et al., 2018*). Our system may be more similar to the continuous improvement observed in T cell activation as our most spaced ligands (36.5 nm) are capable of activating some phagocytosis, albeit not as potently as the 4T. Interestingly, as the intermembrane distance between T cell and target increases, the requirement for tight ligand spacing becomes more stringent (*Cai et al., 2018*). This suggests that IgG bound to tall antigens may be more dependent on tight nanocluster spacing than short antigens. Planar arrays have also been used to vary inter-cluster spacing, in addition to inter-ligand spacing (*Cai et al., 2018*; *Freeman et al., 2016*). Examining the optimal inter-cluster spacing during phagosome closure may be an interesting direction for future studies.

Our study on the spatial requirements of FcγR activation could have implications for the design of therapeutic antibodies or chimeric antigen receptors. Antibody therapies that rely on FcγR engagement are used to treat cancer, autoimmune, and neurodegenerative diseases (*Chao et al., 2010*; *Nimmerjahn and Ravetch, 2005*; *Uchida et al., 2004*; *Watanabe et al., 1999*; *Weiskopf et al., 2013*; *Weiskopf and Weissman, 2015*). Multimerizing Fc domains or targeting multiple antibodies to the same antigen may increase antibody potency (*Zhang et al., 2016*). Interestingly, rituximab, a successful anti-CD20 therapy that potently induces ADCP, has two binding sites on its target antigen (*Zhao et al., 2020*). Selecting clustered antigens or pharmacologically inducing antigen clustering may also increase antibody potency (*Chew et al., 2020*). These results suggest that oligomerization may lead to more effective therapy; however, a systematic study of the spatial parameters that affect FcγR activation has not been undertaken (*Bakalar et al., 2018*). Our data suggest that antibody engineering strategies that optimize spacing of multiple antibodies through leucine zippers, cysteine bonds, DNA hybridization (*Delcassian et al., 2013*; *Seifert et al., 2014*; *Sil et al., 2007*), or multimeric scaffolds (*Divine et al., 2020*; *Fallas et al., 2017*; *Huang et al., 2021*; *Ueda et al., 2020*) could lead to stronger FcγR activation and potentially more effective therapies.

## Materials and methods

**Key resources table**

| Reagent type (species) or resource | Designation | Source or reference | Identifiers | Additional information |
|---|---|---|---|---|
| Antibody | Alexa Fluor 647 anti-biotin IgG (mouse monoclonal) | Jackson Immuno Labs | Cat# 200-602-211 RRID:AB_2339046 | |
| Antibody | Alexa Fluor 488 anti-biotin IgG (mouse monoclonal) | Jackson Immuno Labs | Cat# 200-542-211 RRID:AB_2339040 | |

*Continued on next page*

*Continued*

| Reagent type (species) or resource | Designation | Source or reference | Identifiers | Additional information |
|---|---|---|---|---|
| Sequence-based reagent | Receptor DNA strand | This paper | Benzylguanine-5'-AATATGATGTA TGTGG-3' | Oligonucleotide was ordered from IDT with a 5' terminal amine. Conjugation to benzyl-guanine was performed as described (*Farlow et al., 2013*). |
| Sequence-based reagent | DNA ligand strand | IDT | Biotin-5'-TTTT-TTTC ATACATCATATT-3'-Atto647 | |
| Sequence-based reagent | p8064 DNA scaffold | IDT | Cat# 1081314 | |
| Chemical compound, drug | Alexa Fluor 488 Phalloidin | Thermo/Molecular Probes | Cat# A12379 | |
| Commercial assay, kit | Lipofectamine LTX | ThermoFisher | Cat# 15338030 | |
| Commercial assay, kit | Lenti-X Concentrator | Takara Biosciences | Cat# 631231 | |
| Peptide, recombinant protein | Pierce Biotinylated Bovine Serum Albumin (Biotin-LC-BSA) | ThermoScientific | Cat# 29130 | |
| Peptide, recombinant protein | Neutravidin | ThermoScientific | Cat# 31050 | |
| Cell line (human) | Lenti-X 293 T cell line | Takara Biosciences | Cat# 632180 | For lentivirus production. |
| Cell line (human) | HEK293T cells | UCSF Cell Culture Facility | | For lentivirus production. |
| Cell line (mouse) | Raw264.7 Macrophages | ATCC | Cat# ATCC TIB-71 RRID:CVCL_0493 | |
| Cell line (human) | THP1 Monocytes | ATCC | Cat# ATCC TIB-202 RRID:CVCL_0006 | |
| Transfected construct (mouse) | pHR-DNA-CARγ | This paper | | In PhR vector. Signal peptide: (MQSGTHWRVLGLCLLS VGVWGQD) derived from CD3ε Extracellular: HA tag plus a linker (LPETGGGGGG), SNAPf (from the pSNAPf plasmid, New England Biolabs) Linker: GGSGGSGGS, TM and intracellular: CD86TM (aa 236–271), cytoplasmic domain (aa 45–86) of the Fc γ-chain UniProtKB - P20491 (FCERG_MOUSE) linker: GSGS, Fluorophore: mGFP or BFP. |
| Transfected construct (mouse, human) | pHR-Syk-BFP | Adapted from DOI: 10.1016/j. immuni.2020. 07.008 | | CDS: aa1-629 UniProtKB - P48025 (KSYK_MOUSE), Linker: ADPVAT, Fluorophore: BFP. |

*Continued on next page*

*Continued*

| Reagent type (species) or resource | Designation | Source or reference | Identifiers | Additional information |
|---|---|---|---|---|
| Transfected construct (mouse, human) | pHR-DNA-CARadhesion | DOI: 10.1016/j.immuni.2020.07.008 | | In PhR vector. Signal peptide: (MQSGTHWRVLGLCLLSVGVWGQD) derived from CD3ε Extracellular: HA tag plus a linker (LPETGGGGGG), SNAPf (from the pSNAPf plasmid, New England Biolabs) Linker: GGSGGSGGS, TM and intracellular: CD86TM (aa 236–271), linker: SADASGG, fluorophore: eGFP. |
| Transfected construct (mouse) | pHR-DNA-CARγTM-C25A | This paper | | In PhR vector. Signal peptide: (MQSGTHWRVLGLCLLSVGVWGQD) derived from CD3ε Extracellular: HA tag plus a linker (LPETGGGGGG), SNAPf (from the pSNAPf plasmid, New England Biolabs) Linker: GGSGGSGGS, TM and intracellular: (aa 19–86) of the Fcγ-chain UniProtKB – P20491 (FCERG_MOUSE) with aa25 mutated from C to A linker: GSGS, fluorophore: mGFP or BFP. |
| Transfected construct (mouse) | pHR-DNA-CARγTM-aa26-86 | This paper | | In PhR vector. Signal peptide: (MQSGTHWRVLGLCLLSVGVWGQD) derived from CD3ε Extracellular: HA tag plus a linker (LPETGGGGGG), SNAPf (from the pSNAPf plasmid, New England Biolabs) Linker: GGSGGSGGS, TM and intracellular: (aa 26–86) of the Fcγ-chain UniProtKB – P20491 (FCERG_MOUSE) linker: GSGS, fluorophore: mGFP or BFP. |
| Transfected construct (mouse) | pHR-DNA-CARγ-αTM (aa291–404) | This paper | | In PhR vector. Signal peptide: (MQSGTHWRVLGLCLLSVGVWGQD) derived from CD3ε Extracellular: HA tag plus a linker (LPETGGGGGG), SNAPf (from the pSNAPf plasmid, New England Biolabs) Linker: GGSGGSGGS, TM and intracellular: FcGR1 α-chain (aa 291–404) UniProtKB – P26151 (FCGR1_MOUSE) followed by cytoplasmic domain (aa 45–86) of the Fcγ-chain UniProtKB - P20491 (FCERG_MOUSE) linker: GSGS, fluorophore: mGFP or BFP. |

*Continued on next page*

*Continued*

| Reagent type (species) or resource | Designation | Source or reference | Identifiers | Additional information |
|---|---|---|---|---|
| Transfected construct (mouse, human) | pHR-mNeon Green-tSH2 Syk | Adapted from DOI: 10.1016/j. cell.2018.05.059 | | CDS: aa2-261 UniProtKB - P48025 (KSYK_MOUSE), linker: GGGSGGGG, fluorophore: mNeonGreen. |
| Transfected construct (mouse, human) | pHR-Akt PH domain | This paper | | CDS: aa1–164 UniProtKB – P31749 (AKT1_HUMAN), linker: HMTSPVAT, fluorophore: mGFP. |
| Transfected construct (mouse) | pHR-DNA-CAR4xγ | This paper | | In PhR vector. Signal peptide: (MQSGTHWRVLGLCL LSVGVWGQD) derived from CD3ε Extracellular: HA tag plus a linker (LPETGGGGGG), SNAPf (from the pSNAPf plasmid, New England Biolabs) Linker: GGSGGSGGS, TM and intracellular: CD86TM (aa 236–271), four repeats of the cytoplasmic domain (aa 45–86) of the Fc γ-chain UniProtKB – P20491 (FCERG_MOUSE) with a GSGS linker between each repeat, linker: GSGS, fluorophore: mGFP. |
| Transfected construct (mouse) | pHR-DNA-CAR-N1xγ−3xΔITAM | This paper | | In PhR vector. Signal peptide: (MQSGTHWRVLGLC LLSVGVWGQD) derived from CD3ε Extracellular: HA tag plus a linker (LPETGGGGGG), SNAPf (from the pSNAPf plasmid, New England Biolabs) Linker: GGSGGSGGS, TM and intracellular: CD86TM (aa 236–271), the cytoplasmic domain (aa 45–86) of the Fcγ-chain UniProtKB – P20491 (FCERG_MOUSE) followed by three repeats of the cytoplasmic domain (aa 45–86) of the Fc γ-chain UniProtKB – P20491 (FCERG_MOUSE) with aa65 and aa76 mutated from YtoF and a GSGS linker between each repeat, linker: GSGS, fluorophore: mGFP. |

*Continued on next page*

*Continued*

| Reagent type (species) or resource | Designation | Source or reference | Identifiers | Additional information |
|---|---|---|---|---|
| Transfected construct (mouse) | pHR-DNA-CAR-3xΔITAM-C1xγ | This paper | | In PhR vector. Signal peptide: (MQSGTHWRVLGLCLLSVGVWGQD) derived from CD3ε. Extracellular: HA tag plus a linker (LPETGGGGGG), SNAPf (from the pSNAPf plasmid, New England Biolabs) Linker: GGSGGSGGS, TM and intracellular: CD86TM (aa 236–271), three repeats of the cytoplasmic domain (aa 45–86) of the Fc γ-chain UniProtKB – P20491 (FCERG_MOUSE) with aa65 and aa76 mutated from YtoF and a GSGS linker between each repeat followed by the cytoplasmic domain (aa 45–86) of the Fc γ-chain UniProtKB – P20491 (FCERG_MOUSE), linker: GSGS, fluorophore: mGFP. |
| Transfected construct (mouse) | pHR-DNA-CAR-1xΔITAM-1xγ−2xΔITAM | This paper | | In PhR vector. Signal peptide: (MQSGTHWRVLGLCLLSVGVWGQD) derived from CD3ε. Extracellular: HA tag plus a linker (LPETGGGGGG), SNAPf (from the pSNAPf plasmid, New England Biolabs) Linker: GGSGGSGGS, TM and intracellular: CD86TM (aa 236–271), the cytoplasmic domain (aa 45–86) of the Fc γ-chain UniProtKB – P20491 (FCERG_MOUSE) with aa65 and aa76 mutated from YtoF the cytoplasmic domain, GSGS linker (aa 45–86) of the Fcγ-chain UniProtKB – P20491 (FCERG_MOUSE), GSGS linker, followed by two more repeats of the Fc γ-chain UniProtKB – P20491 (FCERG_MOUSE) with aa65 and aa76 mutated from YtoF the cytoplasmic domain and a GSGS linker between each repeat, linker: GSGS, fluorophore: mGFP. |

*Continued*

| Reagent type (species) or resource | Designation | Source or reference | Identifiers | Additional information |
|---|---|---|---|---|
| Transfected construct (human) | pHR-DNA-CARγ human | This paper | | In PhR vector. Signal peptide: (MQSGTHWRVLGLC LLSVGVWGQD) derived from CD3ε<br>Extracellular: HA tag plus a linker (LPETGGGGGG), SNAPf (from the pSNAPf plasmid, New England Biolabs)<br>Linker: GGSGGSGGS, TM and intracellular: CD86TM (aa 236–271), cytoplasmic domain (aa 45–86) of the Fc γ-chain UniProtKB – P30273 (FCERG_HUMAN) linker: GSGS, fluorophore: mGFP or BFP. |
| Recombinant DNA reagent | pMD2.G lentiviral plasmid | D. Stainier, Max Planck; VSV-G envelope | RRID:addgene_12259 | |
| Recombinant DNA reagent | pCMV-dR8.91 | DOI: 10.1038/ nature11220. | Current RRID:addgene_8455 | |
| Recombinant DNA reagent | pHRSIN-CSGW | DOI: 10.1038/ nature11220. | | |
| Software, algorithm | ImageJ | NIH | | |
| Software, algorithm | Affinity Designer | | | |
| Software, algorithm | Fiji | https://fiji.sc/ | | |
| Software, algorithm | Prism | GraphPad | 8 | |
| Software, algorithm | Micromanager | DOI:10.14440/ jbm.2014.36 | | |
| Other | 5 µm silica microspheres | Bangs | Cat# SS05N | |
| Other | Biotinyl Cap PE | Avanti | Cat# 870273 | |
| Other | POPC | Avanti | Cat# 850457 | |
| Other | PEG5000-PE | Avanti | Cat# 880230 | |
| Other | Atto390 DOPE | ATTO-TEC GmbH | Cat# AD 390-161 | |
| Other | MatriPlate | Brooks | Cat# MGB096-1-2-LG-L | |
| Other | 96-well round bottomed plates | Corning | Cat# 38018 | |
| Other | Illustra NAP-5 columns | Cytiva | Cat# 17085301 | |

## Cell culture

RAW264.7 macrophages were purchased from the ATCC and cultured in DMEM (Gibco, Cat# 11965-092) supplemented with 1× penicillin-streptomycin-L-glutamine (Corning, Cat# 30-009 Cl), 1 mM sodium pyruvate (Gibco, Cat# 11360-070), and 10% heat-inactivated fetal bovine serum (Atlanta Biologicals, Cat# S11150H). THP1 cells were also purchased from the ATCC and cultured in RPMI 1640 Medium (Gibco, Cat# 11875-093) supplemented with 1× Pen-Strep-Glutamine and 10%

heat-inactivated fetal bovine serum. All cells were certified mycoplasma-free and discarded after 20 passages to minimize variation.

## Constructs and antibodies

All relevant information can be found in the Key resources table, including detailed descriptions of the amino acid sequences for all constructs.

## Lentivirus production and infection

Lentiviral infection was used to express constructs described in the Key resources table in either RAW264.7 or THP1 cells. Lentivirus was produced by HEK293T cells or Lenti-X 293 T cells (Takara Biosciences, Cat# 632180) transfected with pMD2.G (a gift from Didier Tronon, Addgene plasmid # 12259 containing the VSV-G envelope protein), pCMV-dR8.91 (since replaced by second generation compatible pCMV-dR8.2, Addgene plasmid #8455), and a lentiviral backbone vector containing the construct of interest (derived from pHRSIN-CSGW, see Key resources table) using lipofectamine LTX (Invitrogen, Cat# 15338-100). The HEK293T media was harvested 60–72 hr post-transfection, filtered through a 0.45 µm filter, and concentrated using Lenti-X (Takara Biosciences, Cat# 631232) via the standard protocol. Concentrated virus was added directly to the cells, and the plate was centrifuged at 2200×g for 45 min at 37°C. Cells were analyzed a minimum of 60 hr later. Cells infected with more than one viral construct were FACs sorted (Sony SH800) before use to enrich for double infected cells.

## DNA origami preparation

The DNA origami pegboard utilized for all experiments was generated as described in *Figure 2—figure supplement 1*. The p8064 DNA scaffold was purchased from IDT (Cat# 1081314). All unmodified oligonucleotides utilized for the origami were purchased from IDT in 96-well plates with standard desalting purification and resuspension at 100 µM in water. Fluorophore and biotin-conjugated oligonucleotides were also purchased from IDT (HPLC purification). All oligonucleotide sequences are listed in *Supplementary file 1*, the assembly is schematized in *Figure 2—figure supplement 1*, and the Cadnano strand diagram for the pegboard with 72 medium-affinity ligands is included in *Figure 2—figure supplement 1*. Core staple oligonucleotides (200 nM) (plates 1 and 2), ligand oligonucleotides (200 nM) (plates 3L, 3MA, and 3 HA), biotinylated oligonucleotides (200 nM), DNA scaffold (20 nM final concentration), and fluorophore-labeled oligonucleotides (200 nM final concentration) were mixed in 1× folding buffer (5 mM Tris pH 8.0, 1 mM EDTA, 5 mM NaCl, 20 mM MgCl$_2$). Origami folding reaction was performed in a PCR thermocycler (Bio-Rad MJ Research PTC-240 Tetrad), with initial denaturation at 65°C for 15 min followed by cooling from 60°C to 40°C with a decrease of 1°C/hr. To purify excess oligonucleotides from fully folded DNA origami, the DNA folding reaction was mixed with an equal volume of PEG precipitation buffer (15% (w/v) PEG-8000, 5 mM Tris-Base pH 8.0, 1 mM EDTA, 500 mM NaCl, 20 mM MgCl$_2$) and centrifuged at 16,000× rcf for 25 min at room temperature (RT). The supernatant was removed, and the pellet was resuspended in 1× folding buffer. PEG purification was repeated a second time, and the final pellet was resuspended at the desired concentration in 1× folding buffer and stored at 4°C.

## Preparation of benzylguanine-conjugated DNA oligonucleotides

5′-amine modified (5AmMC6) DNA oligonucleotides were ordered from IDT and diluted in 0.15 M HEPES pH 8.5 to a final concentration of 2 mM. N-hydroxysuccinimide ester (BG-GLA-NHS)-functionalized benzylguanine was purchased from NEB (Cat# S9151S) and freshly reconstituted in DMSO to a final concentration of 83 mM. To functionalize the oligonucleotides with benzylguanine, the two solutions were mixed so that the molar ratio of oligonucleotide-amine:benzylguanine-NHS is 1:50 and the final concentration of HEPES is between 50 mM and 100 mM. The reaction was left on a rotator overnight at RT. To remove excess benzylguanine-NHS ester, the reaction product was purified the next day with illustra NAP-5 Columns (Cytiva, Cat# 17085301), using H$_2$O for elution. The molar concentration of the benzylguanine-conjugated oligonucleotides was determined by measuring the absorbance of the purified reaction at 260 nm with a Nanodrop. This reaction was further condensed with the Savant SpeedVac DNA 130 Integrated Vacuum Concentrator System, resuspended in water to a final concentration of 100 µM, aliquoted, and stored at −20°C until use.

## Functionalization of glass surface with DNA origami

96-well glass-bottom MatriPlates were purchased from Brooks (Cat# MGB096-1-2-LG-L). Before use, plates were incubated in 5% (v/v) Hellmanex III solution (Z805939-1EA; Sigma) overnight, washed extensively with Milli-Q water, dried under the flow of nitrogen gas, and covered with sealing tape (ThermoFisher, Cat# 15036). Wells used for experiment were unsealed, incubated with 200 µL of Biotin-BSA (ThermoFisher, Cat# 29130) at 0.5 mg/mL in PBS pH 7.4 at RT for 2 hr overnight. Wells were washed 6× with PBS pH 7.4 to remove excess BSA and incubated for 30 min at RT with 100 µL neutravidin at 250 µg/mL in PBS pH 7.4 for origami quantification and 50 µg/mL for cellular experiments. Wells were again washed 6× with PBS pH 7.4 supplemented with 20 mM MgCl$_2$ and incubated for 1–2 hr with the desired amount of DNA origami diluted in PBS pH 7.4 with 20 mM MgCl$_2$ and 0.1% BSA.

## DNA origami quantification

Five wells of a 96-well glass-bottom MatriPlate per origami reaction were prepared as described in 'Functionalization of glass surface with DNA origami'. The purified DNA origami reaction was serially diluted into PBS pH 7.4 with 20 mM MgCl$_2$ and 0.1% BSA, and five different concentrations were plated and incubated for 1.5 hr before washing 5× with PBS pH 7.4 with 20 mM MgCl$_2$ and 0.1% BSA. Fluorescent TIRF images were acquired in the channel with which the origami was labeled. 100 sites per well were imaged using the High Content Screening (HCS) Site Generator plug-in in µManager (*Stuurman et al., 2010*). The number of individual DNA origami per µm$^2$ in each well was quantified using the Spot Counter plug-in in Fiji. This was repeated for all concentrations of origami plated. The final concentration of the origami reaction was measured as number of origami/µm$^2$ and was calculated from a linear fit including all concentrations in which individual origami could be identified by the plug-in.

## TIRF imaging

96-well glass-bottom MatriPlates were functionalized with DNA origami as described and then washed into engulfment imaging media (20 mM HEPES pH 7.4, 135 mM NaCl, 4 mM KCl, 1 mM CaCl$_2$, 10 mM glucose) containing 20 mM MgCl$_2$. Approximately 100,000 dual-infected mNeon-Green-DNA-CARγ and BFP-Syk THP1 cells per well were pelleted via centrifugation, washed into engulfment imaging media, re-pelleted, and resuspended into 50 µL of engulfment imaging media. 1 µL of 100 µM benzylguanine-labeled-receptor DNA stock was added per ~50,000 cells pelleted, and the cell-DNA mixture was incubated at RT for 15 min. Cells were subsequently washed twice via centrifugation with 10 mL of imaging buffer to remove excess benzylguanine-labeled DNA and resuspended in 200 µL per 100,000 cells of imaging buffer containing 20 mM MgCl$_2$. Cells were then immediately added to each well and imaged. Data was only collected from a central region of interest (ROI) in the TIRF field. The origami fluorescent intensities along the x and y axes were plotted to ensure there was no drop off in signal and thus no uniformity of illumination.

## Quantification of receptor and Syk recruitment to individual origami

Cells that expressed both the mNeonGreen-tagged DNA-CARγ receptor and the BFP-tagged Syk and had interactions with the 72-ligand origami were chosen for analysis in Fiji. An ROI was drawn around the perimeter of the cell-glass surface interaction, which was determined by the presence of receptor fluorescence. The 'Spot Intensity in All Channel' plug-in in Fiji (https://github.com/nicost/spotIntensityAnalysis/; *Stuurman, 2020*) was used to identify individual origami pegboards, measure fluorescence intensity of the DNA-CARγ receptor and Syk at each origami pegboard, and subtract local background fluorescence. The intensity at each origami pegboard was normalized to the average intensity measured at 72-ligand origami pegboards in each well.

## Supported lipid bilayer-coated silica bead preparation

Chloroform-suspended lipids were mixed in the following molar ratios: 96.8% POPC (Avanti, Cat# 850457), 2.5% biotinyl cap PE (Avanti, Cat# 870273), 0.5% PEG5000-PE (Avanti, Cat# 880230), and 0.2% atto390-DOPE (ATTO-TEC GmbH, Cat# AD 390-161) for labeled lipid bilayers, or 97% POPC, 2.5% biotinyl cap PE, and 0.5% PEG5000-PE for unlabeled lipid bilayers. The lipid mixes were dried under argon gas and desiccated overnight to remove chloroform. The dried lipids were resuspended

in 1 mL PBS, pH 7.2 (Gibco, Cat# 20012050) and stored under argon gas. Lipids were formed into small unilamellar vesicles via $\geq$30 rounds of freeze-thaws and cleared via ultracentrifugation (TLA120.1 rotor, 35,000 rpm/53,227$\times$g, 35 min, 4°C). Lipids were stored at 4°C under argon gas in an Eppendorf tube for up to 2 weeks. To form bilayers on beads, $8.6 \times 10^8$ silica beads with a 4.89 µm diameter (10 µL of 10% solids, Bangs Labs, Cat# SS05N) were washed 2$\times$ with water followed by 2$\times$ with PBS by spinning at 300 rcf and decanting. Beads were then mixed with 1 mM SUVs in PBS, vortexed for 10 s at medium speed, covered in foil, and incubated in an end-over-end rotator at RT for 0.5–2 hr to allow bilayers to form over the beads. The beads were then washed 3$\times$ in PBS to remove excess SUVs and resuspended in 100 µL of 0.2% casein (Sigma, Cat# C5890) in PBS for 15 min at RT to block nonspecific binding. Neutravidin (Thermo, Cat# 31000) was added to the beads at a final concentration of 1 µg/mL for 20–30 min, and the beads were subsequently washed 3$\times$ in PBS with 0.2% casein and 20 mM $MgCl_2$ to remove unbound neutravidin. The indicated amounts of biotinylated ssDNA or saturating amounts of DNA origami pegboards were added to the beads and incubated for 1 hr at RT with end-over-end mixing to allow for coupling. Beads were washed two times and resuspended in 100 µL PBS with 0.2% casein and 20 mM $MgCl_2$ to remove uncoupled origami pegboards or ssDNA. When functionalizing SUV-coated beads with anti-biotin Alexa Fluor 647-IgG (Jackson ImmunoResearch Laboratories Cat# 200-602-211, Lot# 137445), the IgG was added to the beads at 1 µM immediately following the casein blocking step, and beads were incubated for 1 hr at RT with end-over-end mixing.

## Quantification of ssDNA, IgG, or origami on beads

To estimate the amount of ssDNA bound to each bead, we compared the fluorescence of Atto647-labeled DNA on the bead surface to calibrated fluorescent beads (Quantum Alexa Fluor 647, Bangs Lab) using confocal microscopy (*Figure 1—figure supplement 1*). To determine saturating conditions of IgG and origami pegboards, we titrated the amount of IgG or origami in the coupling reaction and used confocal microscopy to determine the concentration at which maximum coupling was achieved. A comparable amount of origami pegboard coupling was also confirmed with confocal microscopy for beads used in the same experiment.

## Quantification of engulfment

30,000 RAW264.7 macrophages were plated in one well of a 96-well glass bottom MatriPlate (Brooks, Cat# MGB096-1-2-LG-L) between 12 and 24 hr prior to the experiment. Immediately before adding beads, 100 µL of a 1 µM solution of benzylguanine-conjugated receptor DNA in engulfment imaging media was added, incubated for 10 min at RT, and washed out four times with engulfment imaging media containing 20 mM $MgCl_2$, making sure to leave ~100 µL of media covering the cells between washes, and finally leaving the cells in ~300 µL of media. Approximately $8 \times 10^5$ beads were added to the well and engulfment was allowed to proceed for 45 min in the cell incubator. Cells were fixed with 4% PFA for 10 min and washed into PBS. For *Figures 4C* and *6D*, 10 nM Alexa Fluor 647 anti-biotin IgG (Jackson Immuno Labs, Cat# 200-602-211) diluted into PBS containing 3% BSA was added to each well for 10 min to label non-internalized beads. Wells were subsequently washed three times with PBS. Images were acquired using the HCS Site Generator plug-in in µManager and at least 100 cells were scored for each condition. When quantifying bead engulfment, cells were selected for analysis based on a threshold of GFP fluorescence, which was held constant throughout analysis for each individual experiment. For *Figures 3*, *4,* and *6*, and *Figure 4—figure supplement 1*, the analyzer was blinded during engulfment scoring using the position randomizer plug-in in µManager. For the THP1 cells, ~100,000 cells per condition were spun down, washed into engulfment imaging media, and coupled to benzylguanine-labeled receptor DNA as described under TIRF imaging. Cells were resuspended into 300 µL engulfment imaging media containing 20 mM $MgCl_2$ in an Eppendorf tube, ~$8 \times 10^5$ beads were added to the tube, and the tube was inverted 8$\times$ before plating the solution into a round-bottomed 96-well plate (Corning, Cat# 38018). Engulfment was allowed to proceed for 45 min in the cell incubator before the plate was briefly spun and the cells were fixed in 4% PFA for 10 min. Cells were subsequently washed 3$\times$ with PBS by briefly centrifuging the plate and removing the media, and finally moved into a 96-well glass-bottom MatriPlate for imaging.

## Quantification of engulfment kinetics

RAW264.7 macrophages were plated and prepared in wells of a 96-well glass bottom MatriPlate as described in 'Quantification of engulfment'. Using Multi-Dimensional Acquisition in μManager, four positions in the well were marked for imaging at 20 s intervals through at least seven z-planes. Approximately $4 \times 10^5$ Atto647N-labeled 4S origami functionalized beads and ~$4 \times 10^5$ Atto550N-labeled 4T origami functionalized beads were mixed in an Eppendorf tube, added to the well, and immediately imaged. Bead contacts were identified by counting the number of beads that came into contact with the cells throughout the imaging time. Initiation events were identified by active membrane extension events around the bead. Engulfment completion was identified by complete internalization of the bead by the macrophage. The initiation time was quantified as the amount of time between bead contact (the first frame in which the bead contacted the macrophage) and engulfment initiation (the first frame in which membrane extension around the bead was visualized) and was only measured for beads that were completely internalized by the end of the imaging time. The engulfment time was quantified as the amount of time between engulfment initiation and engulfment completion (the first frame in which the bead has been fully internalized by the cell).

## Quantification of synapse intensity of DNA-CARγ receptor, tSH2 Syk, PIP$_3$ reporter, and actin filaments

Phagocytic cups were selected for analysis based on clear initiation of membrane extension around the bead visualized by GFP fluorescence from the DNA-CARγ receptor. The phagocytic cup and the cell cortex (areas indicated in schematic in *Figure 6B*) were traced with a line (six pixels wide for DNA-CARγ receptor and the tSH2 Syk reporter, and eight pixels wide for the Akt-PH reporter and phalloidin) at the Z-slice with the clearest cross section of the cup.

## Microscopy and analysis

Images were acquired on a spinning disc confocal microscope (Nikon Ti-Eclipse inverted microscope with a Yokogawa CSU-X spinning disk unit and an Andor iXon EM-CCD camera) equipped with a $40 \times 0.95$ NA air and a $100 \times 1.49$ NA oil immersion objective. The microscope was controlled using μManager. For TIRF imaging, images were acquired on the same microscope with a motorized TIRF arm using a Hamamatsu Flash 4.0 camera and the $100 \times 1.49$ NA oil immersion objective.

## Statistics

Statistical analysis was performed in Prism 8 (GraphPad, Inc). The statistical test used is indicated in each relevant figure legend.

# Acknowledgements

 We thank N Stuurman for help with microscopy and developing the 'image randomizer' plug-in for blinding our analysis as well as the 'Spot Intensity in All Channel' plug-in for quantification of our TIRF experiments. We also thank K McKinley, T Skokan, C Gladkova, and J Sheu-Gruttadauria for discussions and critical feedback on this manuscript. MAM was supported by the National Institute of General Medical Sciences of the National Institutes of Health under award number F32GM120990. Funding was provided by the Howard Hughes Medical Institute to RDV and the Army Research Office (W911NF-14-1-0507) to SMD.

# Additional information

### Funding

| Funder | Grant reference number | Author |
|---|---|---|
| Howard Hughes Medical Institute | | Ronald D Vale |
| National Institute of General Medical Sciences | F32GM120990 | Meghan A Morrissey |
| Army Research Office | W911NF-14-1-0507 | Shawn M Douglas |

The funders had no role in study design, data collection and interpretation, or the decision to submit the work for publication.

## Author contributions
Nadja Kern, Conceptualization, Resources, Data curation, Formal analysis, Investigation, Visualization, Methodology, Writing - original draft, Writing - review and editing; Rui Dong, Resources, Methodology, Writing - review and editing; Shawn M Douglas, Resources, Visualization, Methodology, Writing - review and editing; Ronald D Vale, Conceptualization, Resources, Supervision, Funding acquisition, Writing - review and editing; Meghan A Morrissey, Conceptualization, Resources, Supervision, Funding acquisition, Writing - original draft, Writing - review and editing

## Author ORCIDs
Nadja Kern [ID] https://orcid.org/0000-0002-1313-5890
Rui Dong [ID] http://orcid.org/0000-0002-9118-3636
Shawn M Douglas [ID] https://orcid.org/0000-0001-5398-9041
Ronald D Vale [ID] https://orcid.org/0000-0003-3460-2758
Meghan A Morrissey [ID] https://orcid.org/0000-0002-0531-4864

## Decision letter and Author response
Decision letter https://doi.org/10.7554/eLife.68311.sa1
Author response https://doi.org/10.7554/eLife.68311.sa2

# Additional files

## Supplementary files
• Supplementary file 1. Oligonucleotide sequences for DNA origami pegboard assemblies. List of staple and ligand strands used to makeup DNA origami pegboards. Plates 1 and 2 have staple strand sequences, and plate 3 variants have sequences used for no ligand (blue), high-affinity (yellow), or medium-affinity (red) ligands at each position of the pegboard.

• Transparent reporting form

## Data availability
All relevant data is in the paper or source data files.

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
