## [Decision Letter]

[Editors' note: this paper was reviewed by Review Commons.]

**Acceptance summary:**

You have utilised DNA origami and synthetic biology tools to convincingly demonstrate that phagocytosis is dependent upon the nanoscale spacing of ligands. Your DNA origami approach is able to operate in a range of intermolecular spacing that sits in between what can be achieved with protein oligomerization and methods based on arraying of metallic particles on planar substrates, which were the previous state of the art. The improved understanding of the physical requirements for phagocytosis that you have generated may also have applications in immunotherapy.

**Decision letter after peer review:**

Thank you for submitting your article "Tight nanoscale clustering of Fc γ-receptors using DNA origami promotes phagocytosis" for consideration by *eLife*. Your article has been reviewed by 3 peer reviewers at Review Commons, and the evaluation at *eLife* has been overseen by a Reviewing Editor and Carla Rothlin as the Senior Editor.

This is a very nicely done synthetic biology/biophysics study on the effect of ligands spacing on phagocytosis. They use a DNA based recognition system similar to that previously use to investigate T cell signaling, but express the SNAP tag linked transmembrane receptor in a macrophage cell line and present the ligands using DNA origami mats to control the number and spacing of complementary ligands that are designed to be in the typical range for low or high affinity FcR, a receptor that can trigger phagocytosis. The study offers valuable quantitative data sets that will be of immediate interest to groups working in this area to understand principled of how this class of receptors work, and in the future, for design of synthetic receptors for immunotherapy applications.

Congratulations on your compelling exploration of the role of ligand spacing in phagocytosis using a DNA based chimeric receptor system and DNA origami.

The original reviewers concur that you have addressed their concerns. There is one issue to clarify.

1. One of the reviewers had asked for experiments with a "receptor fusion construct" (see 10.1038/s41467-019-10097-0) where you retain the alpha1-gamma2 architecture of the native FcR, but add the SNAP-DNA complex to the FcgR1 to determine if this changes the potency or spacing threshold. We understand that this might pose a number of challenges and its OK that you didn't do this. However, the new experiments using the FcgRI α TM and the FcR γ TM without the disulfides may need just a bit more explanation to not confuse readers. The reviewer noted that the FcR γ TM based constructs were more potent at phagocytosis, although, as you point out, they retained the trend of the spacing dependence. Even though the disulfides were not included, it's expected that the FcR γ construct will have a tendency to dimerise and that this may make it more potent. If this is the case, it still doesn't quite address the original question as you now have a synthetic receptor with two ligand binding and two signal transduction modules, whereas the "receptor fusion" would have just one ligand binding unit with two signal transduction modules. The nuances of this need to be discussed better and in relation to the configuration of most natural activating immunoreceptors. I hope this is clear and that you can easily address this.

---

## [Author Response]

Overall, we were pleased that the reviewers found our study carefully designed and interesting. We have addressed their comments below.

Reviewer #1 (Evidence, reproducibility and clarity (Required)):The manuscript by Kern, et al., demonstrates that phagocytosis in macrophages is regulated in part by the intermolecular distance of phagocytosis-promoting receptors engaging phagocytic targets. Cells expressing chimeric receptors containing cytosolic domains of Fc receptors (FcR) and defined ligand-binding DNA domains were used to drive phagocytosis of opsonized glass beads coated with complementary DNA ligands of defined spacing and number. These so-called origami ligands allowed manipulation of receptor spacing following engagement, which allowed the demonstration that tight spacing of ligands (7 nm or 3.5 nm) optimized signaling for phagocytosis. The study is carefully performed and convincing. I have a few technical concerns and minor suggestions.1. It is assumed that the origami preparations were entirely uniform. How much variation was there? Is that supported by TIRF microscopy of origami preparations? Was the TIRF microscopy calibrated for uniformity of fluorescence (ie., shade correction)?

Our laboratory, Dong et al., has extensively characterized the origami uniformity and robustness of these exact pegboards. This paper was just posted on bioRxiv (Dong et. al, 2021). We have also cited this paper in our revised manuscript in reference to the characterization of the DNA origami (Line 117).

We did not use any shade correction. Instead we only collected data from a central ROI in our TIRF field. To check for uniformity of illumination, we plotted the origami pegboard fluorescent intensity along the x and y axis. We observed very modest drop off in signal – the average signal intensity of origamis within 100 pixels of the edge is 76 ± 6% the intensity of origamis in a 100 pixel square in the center of the ROI. Fitting this data with a Gaussian model resulted in very poor R values. While this may account for some of the variation in signal intensity at individual points, we expect the normalized averages of each condition to be unaffected. We have amended the methods to describe this strategy (Lines 851-854).

2. Likewise, how much variation was there in the expression of the chimeric receptors? Large variation in receptor numbers per cell could significantly alter the quantitative studies. Aside from the flow sorting for cells expressing two different molecules, how were cells selected for analysis?

We thank the reviewer for bringing up this point. We confirmed comparable receptor expression levels at the cell cortex of the DNA CAR-𝛾 and the DNA CAR-adhesion used throughout the paper. We also have confirmed that receptor levels at the cell cortex were similar for the large DNA CAR constructs used in Figure 6C-D. This data is now included in Figures S5 and S7. We have also altered the text to include this (lines 169-172):

“Expression of the various DNA CARs at the cell cortex was comparable, and engulfment of beads functionalized with both the 4T and the 4S origami platforms was dependent on the Fc𝛾R signaling domain (Figure S5).”

When quantifying bead engulfment, cells were selected for analysis based on a threshold of GFP fluorescence, which was held constant throughout analysis for each individual experiment. We have amended the “Quantification of engulfment” methods section to convey this ( lines 921-923).

3. The scale of the origami relative to the cells is difficult to discern in Figures 2C and D. Additional text would be helpful to indicate, for example, that the spots on the Figure 2D inset indicate entire origami rather than ligand spots on individual origami particles.

Thank you for pointing this out, we see how the legend was unclear and have corrected it (lines 453-454), including specifically noting “Each diffraction limited magenta spot represents an origami pegboard.” We have also outlined the cell boundary in yellow to make the cell size more clear.

4. Figure 5 legend, line 482: How was macrophage membrane visualized for these measurements?

We have added the following clarification (line 535-536): “The macrophage membrane was visualized using the DNA CAR𝛾, which was present throughout the cell cortex.”

5. Line 265: "our data suggest that there may be a local density-dependent trigger for receptor phosphorylation and downstream signaling". This threshold-dependent trigger response was also indicated in the study of Zhang, et al. 2010. PNAS.

The Zhang et al. study was influential in our study design, and we wish to give the appropriate credit. Zhang et al. found that a sufficient amount of IgG is necessary to activate late (but not early) steps in the phagocytic signaling pathway. In contrast, our study addresses IgG concentration in small nanoclusters. We find that this nanoscale density affects receptor phosphorylation. Thus, we think these two studies are distinct and complementary. Lines 283-287 now read:

“While this model has largely fallen out of favor, more recent studies have found that a critical IgG threshold is needed to activate the final stages of phagocytosis (Zhang et al., 2010). Our data suggest that there may also be a nanoscale density-dependent trigger for receptor phosphorylation and downstream signaling.”

6. Line 55: Rephrase, “we found that a minimum threshold of 8 ligands per cluster maximized FcgR-driven engulfment.” It is difficult to picture how a minimum threshold maximizes something.

We now state “we found that 8 or more ligands per cluster maximized FcgR-driven engulfment.”

7. Line 184: Rephrase, "we created… pegboards with very high-affinity DNA ligands that are predicted not to dissociate on a time scale of >7 hr". Remove "not".

Thank you for pointing this out, it is now correct.

Reviewer #1 (Significance (Required)):This study provides a significant advance in understanding about the molecular mechanisms of signaling for particle ingestion by phagocytosis.Reviewer #2 (Evidence, reproducibility and clarity (Required)):The manuscript on “Tight nanoscale clustering of Fcg-receptors using DNA origami promotes phagocytosis" studies how clustering and nanoscale spacing of ligand molecules for a chimeric Fcg-receptors influence the phagocytosis of functionalized silicon beads by macrophage cell lines. The basis of this study is the design of a chimeric Fc-receptor (DNA-CARg) comprising an extracellular SNAP-tag domain that can be loaded with single-stranded (ss) DNA, the transmembrane part of CD86 and the cytosolic part of the Fc-receptor g-chain containing an immunoreceptor tyrosine-based activation motif (ITAM) as well as a C-terminal green fluorescent protein (GFP). As control the authors used a similar designed DNA-CAR that is lacking the intracellular ITAM-containing FCg tail. The chosen target for this chimeric DNA-CAR, are silicon beads covered by a lipid bilayer that contains biotin-labelled lipids that, via Neutravidin, can be loaded with a biotinylated DNA origami pegboard displaying complimentary ss-DNA as ligand for the DNA-CAR. The DNA origami pegboard contains four ATTO647N fluorescence for visualization and the ssDNA ligand in different quantities and spacing.Using these principles, the authors study how ligand affinity, concentration and spacing influence the activation of the DNA-CARg and the engulfment of the loaded beads.The authors show that bead engulfment is increased between 2 till 8 ssDNA ligands on the pegboard. After this, ligand numbers do not play a role anymore in the engulfment. They then study the role of the ligand spacing using pegboards that either contain 4 single strand DNA ligands in close (7nm/3,5nm) proximity or a more spaced version using 21/17,5 nm or 35/38,5 nm. The authors find that the bead engulfment is maximally and positively affected by the close spacing of the ssDNA ligands. In their final experiments the authors vary the design of the DNA-CARs by tetramerization of the ITAM-containing Fcg-signaling subunit. In their discussion the authors mention different possibilities for the effect of spacing on the engulfment process.I think that, in general, this is an interesting study. However, it has some caveats and open issues that should be clarified before its publication.Major comments1. As a general comment, it is somewhat a pity that the authors did not use the endogenous FcR as a control. It would have been quite easy for the authors to place the SNAP-tag domain on the Fcg extracellular domain which would allow to do all their experiments in parallel, not only with the DNA-CAR, but also with a DNA-containing wild type receptor. Such a control would be important because, by using a CD86 transmembrane domain, the authors do not know whether the nanoscale localization of their chimeric receptors is reflecting that of the endogenous Fcg receptor.

We agree with the reviewer completely. We have repeated experiments shown in Figure 4A with a DNA-CAR containing the Fc𝛾 transmembrane domain instead of CD86 as the reviewer suggests. We also included a DNA-CAR version of the Fc𝛾R1 α chain, although this construct was not expressed as well as the others. These data are now included in Figure S5, and referenced in  lines 167-168.

2. An important issue that is discussed by the authors but not addressed in this manuscript is whether the different amount and spacing of the ligand is only impacting on signaling or also on the mechanical stress of the cells. Indeed, mechanical stress on the cytoskeleton arrangement could influence the engulfment process. For this, it would be very important to test that the different bead engulfment, for example, those shown in Figure 4, is strictly dependent on signaling kinases. The authors should repeat the experiment of Figure 4 a and b in the presence or absence of kinase inhibitors such as the Syk inhibitor R406 or the Src inhibitor PP2 to show whether the different phase of engulfment is dependent on the signaling function of these kinases. This crucial experiment is clearly missing from their study.

We agree this is an interesting point. We find that ligand spacing affects receptor phosphorylation; however this does not preclude effects on downstream aspects of the signaling pathway. We will clarify this by adding the following comment to the manuscript (line 299-301): “While our data pinpoints a role for ligand spacing in regulating receptor phosphorylation, it is possible that later steps in the phagocytic signaling pathway are also directly affected by ligand spacing.”

The DNA-CAR-adhesion in Figure 1 strongly suggests that intracellular signaling is essential for phagocytosis. We have now included additional controls using this construct as detailed in our response to point 3 below. Unfortunately, Src and Syk inhibitors or knockout abrogate Fc𝛾R mediated phagocytosis (for example, PMIDs 11698501, 9632805, 12176909, 15136586) and thus would eliminate phagocytosis in both the 4T and 4S conditions. This precludes analysis of downstream steps in the phagocytic signaling pathway.

3. Another problem of this study is that the authors show in Figure 1A the control DNA-CAR-adhesion but then hardly use it in their study. For example, the crucial experiments shown in Figure 4 should be conducted in parallel with DNA-CAR-adhesion expressing macrophage cells. This study could provide another indication whether or not ITAM signaling is important for the engulfment process.

We have added this control. It is now included in Figure S5 and S7. Figure 3D also shows that the DNA-CAR-adhesion combined with the 4T origami pegboards does not activate phagocytosis and we have amended the text to make this more clear (line 152).

4. Another important aspect is how the concentration of the loaded origami pegboard is influencing the engulfment process. In particular, it would be interesting to show the padlocks with different spacings such as the 4T closed spacing versus 4s large spacing show a different dependency on the concentration of this padlock loading on the beads. This would be another important experiment to add to their study.

We agree that this is an interesting question. We suspect that at a very high origami density, 4S signaling would improve, and potentially approach the 4T. However, we are currently coating the beads in saturating levels of origami pegboards. Thus we cannot increase origami pegboard density and address this directly.

Minor comments:1. The definition of the ITAM is Immunoreceptor Tyrosine-based Activation Motif and not "Immune Tyrosine Activation Motif" as stated by the authors.

We have corrected this.

2. The authors discuss that it is the segregation of the inhibitory phosphatase CD45 from the clustered Fc receptors is the major mechanism explaining their finding that 4T closed spacing is more effective than 4s large spacing. With the event of the CRISPR/Cas9 technology it is trivial to delete the CD45 gene in the genome of the RAW264.7 macrophage cell line used in this study and I am puzzled why they author are not conducting such a simple but for their study very important experiment (it takes only 1-2 month to get the results).

This experiment may be informative but we have two concerns about its feasibility. First, CD45 is a phosphatase with many different roles in macrophage biology, including activating Src family kinases by dephosphorylating inhibitory phosphorylation sites (PMID 8175795, 18249142, 12414720). Second, CD45 is not the only bulky phosphatase segregated from receptor nanoclusters. For example, CD148 is also excluded from the phagocytic synapse (PMID 21525931). CD45 and CD148 double knockout macrophages show hyperphosphorylation of the inhibitory tyrosine on Src family kinases, severe inhibition of phagocytosis, and an overall decrease in tyrosine phosphorylation (PMID 18249142). CD45 knockout alone showed mild phenotypes in macrophages. We anticipate that knocking out CD45 alone would have little effect, and knocking out both of these phosphatases would preclude analysis of phagocytosis. Because of our feasibility concerns and the lengthy timeline for this experiment, we believe this is outside of the scope of our study.

In our discussion, we simplistically described our possible models in terms of CD45 exclusion, as the mechanisms of CD45 exclusion have been well characterized. This was an error and we have amended our discussion to read (lines 335-343):

“As an alternative model, a denser cluster of ligated receptors may enhance the steric exclusion of the bulky transmembrane proteins like the phosphatases CD45 and CD148 (Bakalar et al., 2018; Goodridge et al., 2012; Zhu, Brdicka, Katsumoto, Lin, and Weiss, 2008).”

Reviewer #2 (Significance (Required)):The innovative part of this study is the combination of SNAP-tag attached, chimeric Fc-receptor with the DNA origami pegboard technology to address important open question on receptor function.Referees cross-commentingI find most of my three reviewing colleagues reasonableI also agree to Reviewer #1 comments 2Likewise, how much variation was there in the expression of the chimeric receptors? Large variation in receptor numbers per cell could significantly alter the quantitative studies. Aside from the flow sorting for cells expressing two different molecules, how were cells selected for analysis?But I want to add it is not only the amount of receptors but ils the nanoscale location that is key to receptor function

We have ensured that all receptors are trafficked to the cell surface. We have also measured their intensity at the cell cortex as discussed in response to Reviewer 1.

Reviewer #3 (Evidence, reproducibility and clarity (Required)):This is a very nicely done synthetic biology/biophysics study on the effect of ligands spacing on phagocytosis. They use a DNA based recognition system that the group has previously use to investigate T cell signaling, but express the SNAP tag linked transmembrane receptor in a macrophage cell line and present the ligands using DNA origami mats to control the number and spacing of complementary ligands that are designed to be in the typical range for low or high affinity FcR, a receptor that can trigger phagocytosis. The study offers some very nice quantitative data sets that will be of immediate interest to groups working in this area and, in the future, for design of synthetic receptors for immunotherapy applications. Other groups are working on similar platform for TCR. I don't feel there is any need for more experiments, but I have some questions and suggestions. Answering and considering these could clarify the new biological knowledge gained.

We thank the reviewer for their support of our manuscript. Given the reviewer’s statement that no new experiments are required, we have answered their questions to the best of our ability given the current data. Should the editor decide that any of these topics require experimental data to enhance the significance of the paper, we are happy to discuss new experiments.

Reviewer #3 (Significance (Required)):I think the significance would be increased by addressing these questions, that would help understand how the synthesis system described related to other system directed as similar questions and more natural settings.1. The densities of the freely mobile DNA ligands required to trigger phagocytosis is quite high. Was the length of the DNA duplexes optimized? The entire complex for both the intermediate and high affinity duplexes seems quite short, perhaps <10 nm. Might the stimulation be more efficient if a short stretch of DS DNA is added to increase the length to 12-13 nm?

The extracellular domain of the DNA-CAR (SNAP tag and ssDNA strand) are approximately 10 nm (PMID 28340336). The biotinylated ligand ssDNA is attached to the bilayer via neutravidin, resulting in a predicted 14 nm intermembrane spacing. The endogenous IgG FcR complex is 11.5 nm. Bakalar et al. (PMID 29958103) tested the effect of antigen height on phagocytosis and found that the shortest intermembrane distance tested (approximately 15 nm) was the most effective. As the reviewer notes, the optimal distance between macrophage and target may be larger than our DNA-CAR. However we think the intermembrane spacing in our system is within the biologically relevant range.

We saw robust phagocytosis at 300 molecules/micron of ssDNA, which is similar to the IgG density used on supported lipid bilayer-coated beads in other phagocytosis studies (PMID 29958103, 32768386). As the reviewer noticed, this is significantly higher than ligand density necessary to activate T cells (PMID 28340336). We have added a comment on ligand density to lines 96-97.

2. Are the origami mats generally laterally mobile on the bilayers. If so, what is the diffusion coefficient? Can one detect the mats accumulating in the initial interface between the bead and cell, particularly in cased where there is no phagocytosis? Would immobility of the mats make them more efficient at mediating phagocytosis compared to the monodispersed ligands, which I assume are highly mobile and might even be "slippery".

We have confirmed that our bead protocol generally produces mobile bilayers, where his-tagged proteins can freely diffuse to the cell-bead interface (see accumulation of a his-tagged FRB binding to a transmembrane FKBP receptor at the cell-bead synapse below). We can qualitatively say that the origamis appear mobile on a planar lipid bilayer (see Dong et. al 2021 and images below). Directly measuring the diffusion coefficient on the beads is extremely difficult because the beads themselves are mobile (both diffusing and rotating), and cannot be imaged via TIRF. We do not see much accumulation of the origami at cell-bead synapses. This could reflect lower mobility of the origamis, or could be because the relative enrichment of origamis is difficult to detect over the signal from unligated origamis.

Overall, we expect the origami pegboards (tethered by 12 neutravidins) are less mobile than single strand DNA (tethered by a single neutravidin, supported by qualitative images below). We are uncertain whether this promotes phagocytosis. At least one study suggests that increased IgG mobility promotes phagocytosis (PMID 25771017). However, the zipper model would suggest that tethered ligands may provide a better foothold for the macrophage as it zippers the phagosome closed (PMID 14732161). Hypothetically, ligand mobility could affect signaling in two ways – first by promoting nanocluster formation, and second by serving as a stable platform for signaling as the phagosome closes. Since our system has pre-formed nanoclusters, the effect of ligand mobility may be quite different than in the endogenous setting.

In Author response image 2, a 10xHis-FRB labeled with AlexaFluor647 was conjugated to Ni-chelating lipids in the bead supported lipid bilayer. The macrophages express a synthetic receptor containing an extracellular FKBP and an intracellular GFP. Upon addition of rapamycin, FRB and FKBP form a high affinity dimer, and FRB accumulates at the bead-macrophage contact sites.

**Author response image 2. respfig2:** 

In Author response image 3, single molecules were imaged for 3 sec. The tracks of each molecule are depicted by lines, colored to distinguish between individual molecules. The scale bar represents5 microns in both panels.

**Author response image 3. respfig3:** 

3. Breaking down the analysis into initiation and completion is interesting. When using the non-signalling adhesion constructs, would they get to the initiation stage or would that attachment be less extensive than the initiation phase.

This is an interesting question. While we did not include the DNA-CAR-adhesion in our kinetic experiments, we have now quantified the frequency of cups that would match our ‘initiation’ criteria in 3 representative data sets where macrophages were fixed after 45 minutes of interaction with origami pegboard-coated beads. We found that an average of 16/125 of 4T beads touching DNA-CAR-adhesion macrophages met the ‘initiation’ criteria and an average of 2/125 were eaten (14% total). In comparison, we examined 4T beads touching DNA CAR𝛾 macrophages and found that on average 23/125 met the ‘initiation’ criteria, and 45/125 were already engulfed (54%). This suggests that the DNA-CAR-adhesion alone may induce enough interaction to meet our initiation criteria, but without active signaling from the FcR this extensive interaction is rare. We have added this data in a new Figure S6 and commented on this in lines 213-215.

4. It would be interesting to put these results in perspective of earier work on spacing with planar nanoarrays, although these can't be applied to beads. For integrin mediated adhesion there was a very distinct threshold for RGD ligand spacing that could be related to the size of some integrin-cytoskeletal linkers (PMID: 15067875). On the other hand, T cell activation seemed more continuous with changes in spacing over a wide range with no discrete threshold (PMID: 24117051, 24125583) unless the spacing was increased to allow access to CD45, in which case a more discrete threshold was generated (PMID: 29713075). The results here for phagocytosis with the very small ligands that would likely exclude CD45 seems to be more of a continuum without a discrete threshold, although high densities of ligand are needed. This issue of continuous sensing vs sharp threshold is biologically interesting so would be good assess this by as consistent standards are possible across systems.

We agree that this is an interesting body of literature worth adding to our discussion. We have added a paragraph that puts our study in the context of prior work on related systems, including these nanolithography studies (Line 364-382):

“How does the spacing requirements for Fc𝛾R nanoclusters compare to other signaling systems? Engineered multivalent Fc oligomers revealed that IgE ligand geometry alters Fcε receptor signaling in mast cells (Sil, Lee, Luo, Holowka, and Baird, 2007). DNA origami nanoparticles and planar nanolithography arrays have previously examined optimal inter-ligand distance for the T cell receptor, B cell receptor, NK cell receptor CD16, death receptor Fas, and integrins (Arnold et al., 2004; Berger et al., 2020; Cai et al., 2018; Deeg et al., 2013; Delcassian et al., 2013; Dong et al., 2021; Veneziano et al., 2020). Some systems, like integrin-mediated cell adhesion, appear to have very discrete threshold requirements for ligand spacing while others, like T cell activation, appear to continuously improve with reduced intermolecular spacing (Arnold et al., 2004; Cai et al., 2018). Our system may be more similar to the continuous improvement observed in T cell activation, as our most spaced ligands (36.5 nm) are capable of activating some phagocytosis, albeit not as potently as the 4T. Interestingly, as the intermembrane distance between T cell and target increases, the requirement for tight ligand spacing becomes more stringent (Cai et al., 2018). This suggests that IgG bound to tall antigens may be more dependent on tight nanocluster spacing than short antigens. Planar arrays have also been used to vary inter-cluster spacing, in addition to inter-ligand spacing (Cai et al., 2018; Freeman et al., 2016). Examining the optimal inter-cluster spacing during phagosome closure may be an interesting direction for future studies.”

Additional experiments performed in revision

In addition to these reviewer comments, we have added additional controls validating the DNA-CAR-4x𝛾 used in Figure 6c,d. We compared the DNA-CAR-4x𝛾 to versions of the DNA-CAR-1x𝛾 -3x𝛥 ITAM construct with the functional ITAM in the second and fourth positions (see the schematics now included Figure S7). We found that four individual receptors with a single ITAM each were able to induce phagocytosis regardless of which position the ITAM was in. However the DNA-CAR-4x𝛾 construct, which also contains 4 ITAMs, was not. This further validates the experiment presented in 6c,d. We also fixed minor errors we discovered in the presentation of data for Figures 1C and S1A.

[Editors' note: further revisions were suggested prior to acceptance, as described below.]

The original reviewers concur that you have addressed their concerns. There is one issue to clarify.1. One of the reviewers had asked for experiments with a "receptor fusion construct" (see 10.1038/s41467-019-10097-0) where you retain the alpha1-gamma2 architecture of the native FcR, but add the SNAP-DNA complex to the FcgR1 to determine if this changes the potency or spacing threshold. We understand that this might pose a number of challenges and its OK that you didn't do this. However, the new experiments using the FcgRI α TM and the FcR γ TM without the disulfides may need just a bit more explanation to not confuse readers. The reviewer noted that the FcR γ TM based constructs were more potent at phagocytosis, although, as you point out, they retained the trend of the spacing dependence. Even though the disulfides were not included, its expected that the FcR γ construct will have a tendency to dimerise and that this may make it more potent. If this is the case, it still doesn't quite address the original question as you now have a synthetic receptor with two ligand binding and two signal transduction modules, whereas the "receptor fusion" would have just one ligand binding unit with two signal transduction modules. The nuances of this need to be discussed better and in relation to the configuration of most natural activating immunoreceptors. I hope this is clear and that you can easily address this.

We apologize for misunderstanding this concern in the original comment. We did unsuccessfully try to generate an Fc ⍺ chain DNA CAR. We thought this CAR may be able to recruit the endogenous Fc γ chains and signal with the 1 ligand:2 signaling domain architecture present in most, but not all, activating FcγRs. However, this construct failed to induce engulfment of beads coated in ssDNA. Upon further examination we found that the majority of the protein was not trafficked to the cell surface. The TRuC strategy may be a better approach for generating a receptor with the correct architecture, although the position of the SNAP tag, the expression levels of the modified ⍺ chain, and the assembly into a multimeric receptor would all need to be controlled or optimized to ensure each ligand binding event correlates with a fully assembled complex and the overall architecture of the phagocytic synapse is maintained (ie, intermembrane spacing). Thus, we believe this is outside the scope of our study.

Instead, we have taken the suggestion to clarify the nuances of our study. We have amended the Results section to better describe the CARs with the Fc γ or ⍺ chain transmembrane domains (lines 163-169). We have also added the following paragraph to the discussion clarifying how our DNA CAR compares to the native FcγR complex:

“How does our synthetic DNA-CAR𝛾 receptor compare to endogenous FcγRs? Our DNA CARs are single chain receptors that recruit one intracellular signaling domain per ligand, similar to the single chain human FcγRIIA receptor (Nimmerjahn and Ravetch, 2006). FcγRIIA is ubiquitously expressed on human myeloid cells, and high affinity FcRIIA alleles correlate with an increase in effectiveness of the ADCP-inducing drug Rituximab and lupus susceptibility (Bruhns and Jönsson, 2015; Nimmerjahn and Ravetch, 2006). The majority of FcγR family members, including all activating mouse FcγRs and the human FcγRI and FcγRIIIA are multimeric complexes composed of a ligand binding ⍺ chain and a dimerized signaling 𝛾 chain. This results in two signaling 𝛾 chains recruited to each IgG ligand. The different stoichiometry between ligand binding and intracellular signaling domains may affect some parameters like optimal cluster size. A second difference between the DNA-CAR𝛾 and the endogenous system is the presence of the CD86 transmembrane domain. We found that ligand spacing had a similar effect on phagocytosis when we replaced the CD86 transmembrane domain with the Fc ⍺ or Fc γ transmembrane domain. However, the Fc γ transmembrane domain construct triggered more bead internalization across all conditions. We hypothesize this could be because the transmembrane domain retains some ability to dimerize, recruiting more signaling domains to each ligand, or because it is better able to associate with lipid-ordered domains. Future studies that pattern either endogenous Fc Receptor complex or IgG ligand could clarify these questions.”